

# Multimodel simulation of vertical gas transfer in a temperate lake

Sofya Guseva[1], Tobias Bleninger[2], Klaus Jöhnk[3], Bruna Arcie Polli[2], Zeli Tan[4], Wim Thiery[5,6], Qianlai Zhuang[7], James Anthony Rusak[8], Huaxia Yao[8], Andreas Lorke[1], and Victor Stepanenko[9,10]

[1]Institute for Environmental Sciences, University of Koblenz- Landau, Landau, Germany. *
[2]Graduate Program on Water Resources and Environmental Engineering, Federal University of Paraná , Curitiba, Brazil.
[3]CSIRO Land and Water, Black Mountain, Canberra ACT 2601, Australia.
[4]Pacific Northwest National Laboratory, Richland, Washington, USA.
[5]Institute for Atmospheric and Climate Science, ETH Zurich, Zurich, Switzerland.
[6]Department of Hydrology and Hydraulic Engineering, Vrije Universiteit Brussel, Brussels, Belgium.
[7]Department of Earth, Atmospheric, and Planetary Sciences, Purdue University, West Lafayette, Indiana, USA.
[8]Dorset Environmental Science Centre, Ontario Ministry of Environment, Conservation and Parks, Dorset, Ontario, P0A 1E0, Canada.
[9]Laboratory for Supercomputer Modeling of Climate System Processes, Research Computing Center, Lomonosov Moscow State University, Moscow, Russia.
[10]Department of Meteorology and Climatology, Faculty of Geography, Lomonosov Moscow State University, Moscow, Russia.

*Correspondence to:* Sofya Guseva (guseva@uni-landau.de)

**Abstract.**

In recent decades, several lake models of varying complexity have been developed and incorporated in numerical weather prediction systems and climate models. To foster enhanced forecasting ability and verification, improvement of these lake models remains essential. This especially applies to the limited simulation capabilities of biogeochemical processes in lakes and greenhouse gas exchanges with the atmosphere. Here we present multi-model simulations of physical variables and dis-

solved gas dynamics in a temperate lake (Harp Lake, Canada). The five models (ALBM, FLake, LAKE, LAKEoneD, MTCR-1) considered within this most recent round of the Lake Model Intercomparison Project (LakeMIP) all captured the seasonal temperature variability well. In contrast, none of the models is able to reproduce the exact dates of ice-cover and ice-off, leading to considerable errors in the simulation of eddy diffusivity around those dates. We then conducted an additional modeling experi-

ment with a diffusing passive tracer to isolate the effect of the eddy diffusivity on gas concentration. Remarkably, sophisticated $k - \varepsilon$ models do not demonstrate a significant difference in the vertical diffusion of a passive tracer compared to models with much simpler turbulence closures. All models simulate less intensive spring overturn compared to autumn. Reduced mixing in the models consequently leads to the accumulation of the passive tracer distribution in the water column. The lake models with a comprehensive biogeochemical module, such as ALBM and LAKE, predict dissolved oxygen dynamics adequate to the

observed data. However, for the surface carbon dioxide concentration the correlation between modeled (ALBM, LAKE) and observed data is weak ($\sim$0.3). Overall our results indicate the need to improve the representation of physical and biogeochemical processes in lake models, thereby contributing to enhanced weather prediction and climate projection capabilities.

*Previous affiliation of S.Guseva: Department of Geography, Lomonosov Moscow State University, Moscow, Russia.



# 1 Introduction

The past two decades have seen a renewed interest in lake modeling. Due to the ever-increasing spatial resolution in numerical weather prediction systems and climate models, many lakes became resolvable on the model grid. Lakes are responsible for changing local atmospheric conditions by modifying turbulent fluxes of heat, moisture and momentum relative to adjacent land (Forbes and Meritt, 1984; Mahrt, 2000; Long et al., 2007; Thiery et al., 2015). This effect can be substantial over regions where the total coverage of inland water bodies is high, e.g. in Finland or Sweden ($\approx$ 10 %, 9 % lakes of the total area, respectively (Lehner and Döll, 2004)), or where large lakes dominate (Docquier et al., 2016; Thiery et al., 2015, 2016, 2017). Additionally, lakes are increasingly recognized as a considerable source of greenhouse gases, such as carbon dioxide ($CO_2$) and methane ($CH_4$), to the atmosphere (Tranvik et al., 2009; Bastviken et al., 2011; Raymond et al., 2013; Tan and Zhuang, 2015; Wik et al., 2016).

With the development of multiple lake models of varying complexity, it becomes necessary to compare them and examine their merits and drawbacks in the context of atmospheric and limnological applications. The scientific community has addressed this need through the Lake Model Intercomparison Project (LakeMIP) launched in 2008 (http://netfam.fmi.fi/Lake08/). Since then, a methodology for lake model comparison has been developed (Stepanenko et al., 2010) and lake model intercomparison experiments have been conducted for a range of limnological and climatic conditions. In each experiment, input parameters were unified as much as possible, including identical meteorological forcing, optical properties of water, ice, and snow, initial temperature profiles, and lake depth and/or morphometry. This allowed for a detailed inter-model comparison of process implementations.

Previous LakeMIP studies primarily investigated the ability of the lake models to simulate the thermal regime of the water bodies that cover a wide range of size, depth and mixing regimes in different latitudes. In particular, the major effort has been spent on the simulation of the evolution of the vertical temperature profile along with modeling of the surface temperature and energy fluxes to the atmosphere. All lake model intercomparison studies so far have been limited to the thermal regime of lakes - biogeochemical processes were not considered.

LakeMIP simulations have been performed using seven one-dimensional lake models: 1) CLM4-LISSS (Hostetler and Bartlein, 1990; Subin et al., 2012), 2) FLake (Mironov, 2003), 3) Hostetler model (Hostetler et al., 1993), 4) LAKE (Stepanenko et al., 2011, 2016), 5) LAKEoneD (Jöhnk and Umlauf, 2001; Jöhnk et al., 2008), 6) MINLAKE96/2012 (Fang and Stefan, 1996), and 7) SimStrat (Goudsmit et al., 2002). They targeted deep lakes: monomictic Lake Geneva (Switzerland/France), dimictic Lake Michigan (USA), meromictic equatorial Lake Kivu (Central Africa), and for shallow lakes: Lake Sparkling (USA), Lake Großer Kossenblatter See (Germany), Lake Valkea-Kotinen (Finland). The lake models in general simulated reasonable seasonal variability of temperature and thermocline characteristics (Perroud et al., 2009; Stepanenko et al., 2010). However, the simulated and observed evolution of the mixed layer thickness and the bottom temperature tend to disagree. The $k - \varepsilon$ models (SimStrat, LAKE, LAKEoneD) demonstrate intensive mixing of the lakes during summer, emphasizing their applicability especially to shallow lakes (Stepanenko et al., 2010, 2014). Nevertheless, this type of lake models can also be appropriate for studying physical processes in deep lakes (Thiery et al., 2014a). Furthermore, lake models using the Hendersen-



Sellers parameterisation of vertical mixing (e.g. CLM4-LISSS, Hostetler lake model, (Henderson Sellers, 1985)) represent the thermodynamic state of a water body adequately, but can underestimate mixolimnion temperatures (Thiery et al., 2014a). The FLake model has demonstrated less satisfactory agreement with observations in terms of summer stratification in previous studies, yet simulates surface temperatures well.

This paper presents a logical continuation of LakeMIP by advancing intercomparisons to the study of biogeochemical processes in lakes. It primarily focuses on the simulation of key physical factors affecting the vertical transport of greenhouse gases, such as thermal stratification, vertical diffusion of gases, and ice cover. The ice cover is of special interest for greenhouse gas dynamics, as it can cause the depletion of oxygen, which favors $CH_4$ accumulation until spring ice breakup (Phelps et al., 1998; Karlsson et al., 2013; Jammet et al., 2015).

Our study aims at identifying the major merits and shortcomings of different lake model formulations, as well as uncertainties in simulating the limnophysical controls on greenhouse gas distribution and emissions to the atmosphere. Simulations were run for Harp Lake, a lake in Ontario, Canada with long-term high-frequency monitoring data for meteorology, water temperature, $CO_2$ and $O_2$. Five lake models were used in this study to simulate thermal dynamics and turbulent diffusivity in the lake: 1) Arctic Lake Biogeochemistry Model, ALBM (Tan et al., 2015; Tan and Zhuang, 2015; Tan et al., 2017, 2018) 2) FLake

(Mironov, 2003) 3) LAKE (Stepanenko et al., 2011, 2016) 4) LAKEoneD (Jöhnk and Umlauf, 2001; Jöhnk et al., 2008) 5) Modelagem do Transporte de Calor no Reservatório, MTCR-1 (Polli and Bleninger, 2015, 2018). ALBM and LAKE models include comprehensive biogeochemical modules for calculation of dissolved gas concentrations, which were then tested in their ability to reproduce $CO_2$ and oxygen $O_2$ dynamics. The study is structured as follows: Sect. 2 includes the description of the study site, observational data, lake models, and experimental setup. Section 3 is devoted to the simulation of the lake's thermal

regime, i.e. the temperature patterns and eddy diffusivity (ED). To assess the pure effect of the ED on the gas distribution and dynamics in the lake, additional experiments have been conducted, where the fate of passive tracers emitted from the bottom of the lake was analyzed. In a final section, the models ALBM and LAKE are compared in terms of the $CO_2$ and $O_2$ concentration in Harp Lake.

## 2    Materials and methods

### 2.1    Study site and data

#### 2.1.1    Study area

The object of the current study is Harp Lake ($45°$ $22'$ N , $79°$ $07'$ W, 327 m above sea level), a dimictic temperate lake located in south-central Ontario, Canada. The lake and its surrounding forested catchment are representative of the southern Canadian Shield landscape (Cox, 1978; Dillon et al., 1987) (Fig. 1). It has a 0.71 km$^2$ surface area, a volume of 0.0095 km$^3$, a maximum

depth of 37.5 m and mean depth of 13.32 m. The lake is oligotrophic, characterized by average total phosphorus and chlorohyll-a concentrations of 6.7 and 3.5 µg l$^{-1}$ (Molot and Dillon, 1991). The water is slightly acidic (pH 6.3) (Dillon et al., 1987). Harp Lake has six inflows and one outflow, with a long-time average discharge of 0.067 m$^3$ s$^{-1}$ for the inflows and 0.091 m$^3$





$s^{-1}$ for the outflow. The difference is due to small ungauged inflows and overland flow. Its average residence time is 3.1 years. Due to this relatively long residence time, the effect of the inflows on temperature can be neglected. The lake has only small water level fluctuations (maximum ~0.2 m).

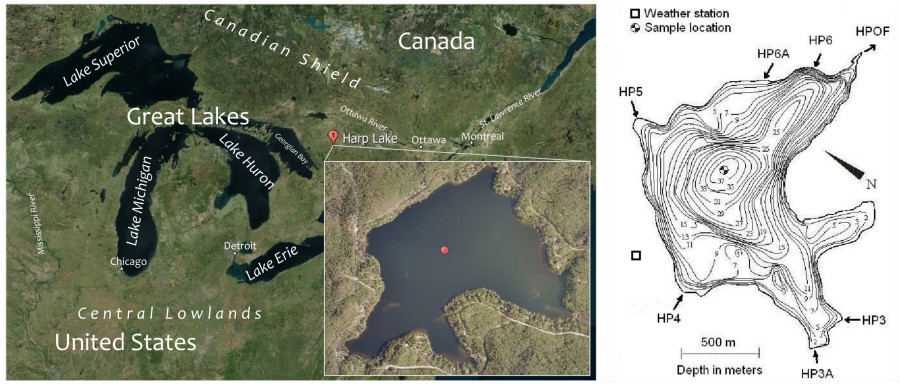

**Figure 1.** The location and morphometry of Harp Lake (HP3 to HP6A are the six inflows, HPOF is the lake outflow (Yao et al., 2014)). Red circle indicates the position of the high-frequency monitoring buoy (45° 22′ 44.88″ N, 79° 8′ 7.77″ W).

### 2.1.2 Observational data

The lake has been monitored by the Dorset Environmental Science Centre (DESC) for more than 40 years, thus providing a comprehensive database supporting the model experiments. The high-frequency and continuous observations used in this study have been collected during 5 years (14.07.2010-19.10.2015) with 10-minute resolution on a monitoring buoy in the lake center (Fig. 1). The meteorological variables collected are: air temperature, humidity at the height of 1.75 m above the water surface; wind speed at 1.75 m, and downwelling shortwave and longwave radiation. Pressure and precipitation are available

from a land-based meteorological station 500 m away from the western lake shore. The high-frequency dataset further includes the a vertical temperature profile (up to the depth of 27.1 m prior to the summer of 2012 and to a depth of 9.85 m thereafter, aggregated to time step of 1 h), $O_2$ concentrations (at two depths: 1 m, 18 m, aggregated to a time step – 1 h) and $CO_2$ concentration (at a depth of 0.39 m, for the period 12.03.2012 – 19.10.2015, aggregated to a time step of 1 h). In addition, traditional temperature and $O_2$ profiles were collected to the depth of 35 m, except during the winter time period, at time step

of approximately 2 weeks, and Secchi depth was also measured fortnightly or monthly during these 5 years (for more details see SI Table  S6,  S7).

   The meteorological conditions of the region during the period under study are typical for continental climate of mid-latitudes (Table 1). This region is subject to the influence of the Azores High during summer, and is located in between the Icelandic Low, and the Canadian High forming in the central part of the continent during winter. Cold arctic air usually invades this area

during winter, leading to the formation of stable freezing weather. However, it should be considered that winters are usually characterized by several abrupt rises in air temperature above 0 °C (see SI, Fig. S1). This phenomenon causes the 2012 winter



to rank as the warmest winter during the 5 measurement years. Average annual wind speed is relatively low ($\approx 2$ m s$^{-1}$), likely caused by wind-shading effects imposed by the surrounding forest. Predominant wind directions are WSW (230° - 280°) and SE (130°).

**Table 1.** Hydrometeorological conditions during the period 14.07.2010 – 19.10.2015.

| | Max | Min | Average | Standard deviation, $\sigma^*$ |
|---|---|---|---|---|
| Air temperature, $T$ °C | 31.4 | -37.5 | winter: -9.2 summer: + 18 | 12.4 |
| Lake surface temperature, $T_s$ °C [1] | 28.4 | -1.5 | 10.9 | 9.42 |
| $\Delta T = T_s - T_{27.1m}$, °C | 24.1 | -4.3 | 5.6 | 9 |
| Wind speed, $W$ m s$^{-1}$ | 9 | 0 | 2 | 1.28 |
| Precipitation, $R$ mm h$^{-1}$ | 39.7 | 0 | 0.12 | 0.71 |

a. [1] Water temperature measured at 0.1 m depth.

## 2.2 Lake models

In this study, a suite of 1-D lake models was applied to Harp Lake. Two classes of finite difference models can be identified based on different approaches of calculating the vertical ED: Henderson-Sellers-based models (Henderson Sellers, 1985; Hostetler and Bartlein, 1990) which are computationally inexpensive, and the more sophisticated models based on $k - \varepsilon$ turbulence closure scheme. The Henderson-Sellers approach computes the ED coefficient as a function of the Richardson number and considers wind-driven turbulence during stable or neutral stratification. A separate procedure is considered for convective mixing and assumes instantaneous mixing of unstably stratified depth layers. The $k - \varepsilon$ formulation involves prognostic equations for turbulent kinetic energy and its dissipation. Both wind-induced and convective turbulence are taken into account in this approach. Within the first class there are recently developed models ALBM and MTCR-1; while $k - \varepsilon$ models include LAKE and LAKEoneD.

The FLake model stands aside from the other 1-D models due to the two-layer bulk structure which employs the concept of self-similarity to estimate the temperature profile in the mixed layer, and thermocline, respectively (Mironov, 2008). In the mixed layer the temperature is assumed to be constant, whereas below it is parameterized as a function of non-dimensional depth. ED is not explicitly specified in this model. Of all models, FLake has the lowest computational cost (Thiery et al., 2014a).

Ice formation is one of the main controlling factors for the dissolved gas dynamics in a lake during winter. Models like LAKEoneD and MTCR-1 involve a simple ice parameterisation, in which ice-cover is governed only by air temperature and lake temperature is not considered (Ashton, 1980, 2011). The FLake model assumes a linear profile of temperature in ice depending on X, Y, Z (Mironov, 2008). The ALBM model incorporates a more comprehensive ice parameterisation developed by Fang and Stefan (1994), taking into account the physical processes of freezing of the surface water, radiation and heat penetration through ice and snow. The LAKE model takes into account the same physical processes and uses a multilayer ice





parameterisation (Stepanenko and Lykossov, 2005; Stepanenko et al., 2011) based on an unsteady heat transfer equation. Light extinction in all models follows the Beer–Lambert law with a constant light attenuation coefficient.

ALBM and LAKE incorporate the most comprehensive biogeochemical modules, including the sources, sinks and transport of $CO_2$, $O_2$, $CH_4$ (and nitrogen in ALBM) (see SI, Table S1). LAKEoneD model calculates $O_2$ concentration only. Parameters
5 and characteristics of all models are summarize in Table 2. All models use finite-difference numerical schemes along vertical coordinates, except for FLake which is based on the two-layer representation of lake temperature profile (see above). The vertical grid of these models is fixed in time, while the number of numerical layers varies from 51 to 105 between models. The time step varies within the range of 20-3600 s between the models. ALBM uses a fourth-order adaptive Runge-Kutta-Fehlberg solver and a variable time step. If there are abrupt changes of boundary conditions (such as air temperature, solar radiation and
10 wind), the time step in this model will be reduced to avoid numerical instability.

**Table 2.** Characteristics of lake models relevant to the study.

| Lake model | Number of layers | Time step (s) | Turbulent mixing parameterisation | Gases considered in biogeochemical module / analyzed in experiments | Ice module |
|---|---|---|---|---|---|
| ALBM | 51 | 1-1800 | Henderson-Sellers-based | $CO_2$, $O_2$, $CH_4$, $N_2$ / $CO_2$, $O_2$ | bulk model (Fang and Stefan, 1994) |
| MTCR-1 | 95 | 3600 | Henderson-Sellers-based | – | simple model (Ashton, 1980, 2011) |
| FLake | 2 | 3600 | two-layer bulk model, assuming self-similar temperature profile | – | bulk model, assuming self-similar temperature profile |
| LAKE | 105 | 20 | $k - \varepsilon$ | $CO_2$, $O_2$, $CH_4$ / $CO_2$, $O_2$ | multilayer model (Stepanenko and Lykossov, 2005) |
| LAKEoneD | 75 | 240 | $k - \varepsilon$ | $O_2$ | simple model (Ashton, 1980, 2011) |



## 2.3 Experimental setup

In this study, a set of numerical experiments were conducted to compare model simulation outputs assuming different morphometries, light extinction coefficients, and for the biogeochemical model parts initial $CO_2$ concentrations (Table 3):

    1) a reference simulation in which the depth is set to a maximum value (37.5 m), light attenuation coefficient is set to a mean

measured value ($\mu$= 0.4 m$^{-1}$) and the morphometry is not taken into account (dependence of horizontal cross-section area on depth (hypsometric curve) is not included). [RefSim];

    2) model simulations testing the sensitivity of models to variations in:

  – lake depth , setting it to mean value: $h_{ave}$ = 13.32 m [LDSim],

  – the light attenuation coefficient, prescribing $\mu_{min}$ = 0.28 m$^{-1}$, and $\mu_{max}$ = 0.68 m$^{-1}$ which are minimal and maximal values

measured, respectively [ExtMinSim, ExtMaxSim],

    3) a model simulation testing the effect of the vertical diffusion on the distribution of the passive tracer [PassTrSim].

    4) a model simulation with biogeochemical modules activated including the calculation of the $O_2$ and $CO_2$ concentration [GasSim]. Measured mean $CO_2$ concentration and $O_2$ observations are used as initial profiles. Time step of model output in all experiments is 1 h.

**Table 3.** Parameters of numerical experiments of LakeMIP-Harp.

| Experiment (14.07.2010 - 19.10.2015) [1] | Light attenuation coefficient, $\mu$, m$^{-1}$ | Lake depth, m | Hypsometric curve | Initial $CO_2$ concentration, mol·m$^{-3}$ | Participating models |
|---|---|---|---|---|---|
| RefSim | 0.4 | 37.5 | – | – | all |
| LDSim | 0.4 | 13.32 | – | – | all |
| ExtMinSim, ExtMaxSim | 0.28/0.68 | 37.5 | – | – | all |
| PassTrSim | 0.4 | 37.5 | – | – | ALBM, MTCR-1, LAKE, LAKEoneD |
| GasSim | 0.4 | 37.5 | + | 0.098 | ALBM, LAKE |

a. [1] For more details see the simulation protocol in SI. Some of the experiments are not included into our study.

The meteorological forcing is identical for all models and experiments. This, however, does not imply that heat fluxes to the atmosphere are the same in models, as models use different surface flux schemes. Some of the models do not include bathymetry (e.g. FLake); thus, for a more consistent comparison, in the RefSim experiment all models are integrated assuming homogeneous depth. In the experiment GasSim, lake morphometry is considered because the lake's carbon budget cannot be accurately calculated without including sediment oxygen demand along a sloping bottom.



The effect of the tributaries is neglected due to the large residence time of Harp Lake. Likewise, water level variations are not considered in the models given the small temporal fluctuations.

The choice of the lake depth in the one-dimensional model without bathymetry may be crucial for the modeling results, not only for the temperature regime (Balsamo et al., 2010) but also for the gas dynamics. The gas dynamics is affected by lake depth as the latter controls the amount of bottom-originated gas to be biochemically-transformed in the water column (e.g. oxidation of $CH_4$). Thus, it is important to examine the sensitivity of models to this parameter. In all experiments except LDSim, the maximal depth is used to compare the output with all available measurements. To test the effect of the chosen depth on model performance, the depth is set to mean depth (13.32 m) in the LDSim experiment, as such the modeled lake volume is equal to the real lake volume.

The light attenuation coefficient is an important parameter, regulating the thermal regime of a lake (Heiskanen et al., 2015) and biogeochemical processes including photosynthesis. Secchi depth measurements during the 5-year study period on Harp Lake are irregular but enable calculation of mean, maximum and minimum light attenuation coefficient using the Poole and Atkins formula (Poole and Atkins, 1929).

The motivation for conducting experiments with perturbed input parameters (depth and light attenuations coefficient) and initial conditions (dissolved gases) is that these properties, especially light attenuation coefficient, become considerably uncertain in global applications. While a first global lake depth database recently became available (Choulga et al., 2014; Kourzeneva, 2010), such data set currently does not exist for water transparency.

To understand the role of ED coefficient in gas distribution and dynamics, an additional experiment solving the pure vertical diffusive transport equation for a passive tracer governed by eddy diffusivities resulting from different lake models was conducted. Details on this experiment are presented in Sec. 3.3.

Harp Lake is a deep lake suitable for studying the transport of $O_2$ and $CO_2$ as well as their biogeochemical transformations due to long-term regular measurements of these gases. In this respect, the additional numerical experiment [GasSim] was conducted with ALBM and LAKE, the lake models with most sophisticated biogeochemical modules. To harmonize the experimental setup, the contribution of $CO_2$ and $O_2$ fluxes through a sloping bottom to the lake carbon budget was switched on in the LAKE model. It should be noted that ALBM model includes $CO_2$ sink/source due to outlets and inlets, whereas LAKE does not take this effect into account. In the ALBM model, a set of biogeochemical parameters was calibrated, specifically: maximum chlorophyll-specific photosynthetic rate, maximum metabolic loss potential, aquatic dissolved organic matter (DOM) microbial degradation rate, terrestrial DOM microbial degradation rate, groundwater dissolved organic carbon (DOC) concentration (Tan et al., 2017). No parameter calibration was performed for the LAKE model (a complete set of default values of model parameters is given in (Stepanenko et al., 2016)).

The model skill scores used in this study are listed in Table 4.





**Table 4.** Statistical measures of model performance.

| Symbol | Property | Definition |
|---|---|---|
| $f_n, r_n$ | Model and observed data | |
| $\sigma_f, \sigma_r$ | Standard deviations of f and r, respectively | |
| DM | Difference between the mean values of the model and the observed data | $DM = \overline{f} - \overline{r}$ |
| R | Correlation coefficient | $R = \dfrac{\frac{1}{N}\sum_{n=1}^{N}(f_n - \overline{f})(r_n - \overline{r})}{\sigma_f \sigma_r}$ |
| $RMSE_c$ | Centered root mean square error | $RMSE_c = \left(\frac{1}{N}\sum_{n=1}^{N}\left[(f_n - \overline{f}) - (r_n - \overline{r})\right]^2\right)^{\frac{1}{2}}$ |





## 3 Results and Discussion

### 3.1 Temperature and ice

All lake models, with the exception of FLake (see below), simulate seasonal water temperature dynamics adequately (Fig. 2):
the epilimnion is warming throughout the first half of summer, accompanied by thermocline deepening, eventually leading to

overturn in autumn. During winter, the lake temperature is at the freezing point immediately below the ice-cover, causing stable
stratification. During spring, the lake starts to mix due to the penetration of shortwave radiation. The ice formation and ice melt
dates play a key role in various processes: the interaction of the lake surface with the atmosphere, the overturning during spring,
biogeochemical processes, release of gases that have accumulated during wintertime, e.g. $CO_2$ or $CH_4$. Yet none of the models
is able to precisely predict these dates. The difference between the model results and observations become even larger towards

the end of simulation, with differences exceeding 2 weeks in winter 2014-2015 (see SI, Table S3).

*FLake model.* The vertical temperature profiles produced by FLake model are notably different from other models. As
highlighted in former studies (e.g. Stepanenko et al. (2010); Thiery et al. (2014b)), this model overestimates the surface mixed
layer depth during summer, even under ice (Fig. 2 (b)). In FLake, the temperature gradient in the thermocline is weak and its
thickness is almost 20-30 m, in contrast to observations showing a thickness of only 5-10 m. This is caused by the shape of the

self-similar temperature profile in the model. The mixed-layer depth in the FLake model is 1.5 times larger than in observed
data (5.6 m versus 3.6 m). Predictions of ice-cover and ice-off are characterized by large errors (see SI, Table S3), with modeled
dates more than two weeks off from the observations. During the warmest winter (2011-2012), FLake does not simulate an ice
layer at all.

FLake demonstrates a considerable error ( $RMSE_c$) in the temperature profile (Fig. 3 (a)), in particular for the depth of

thermocline ($RMSE_c$ up to 5.8 ° C). In contrast, the model performance in simulation of the lake surface temperature is
comparable with other models and the error is 1.5 times smaller (3.73 ° C) (see SI, Fig. S2 (a)), which points to the fact that it
includes similar formulations for the surface heat balance as the other models.

Remarkably, FLake demonstrates the largest sensitivity among all models to the lake depth – $RMSE_c$ is reduced approxi-
mately twofold (from 3.73 ° C to 1.48 ° C) (see SI, Table S4) when using a mean depth of 13.32 m instead of the maximal

depth 37.5 m. This suggests that the model is most suited for shallow water bodies, not as deep as Harp Lake. At the same
time, FLake showed the smallest sensitivity of the surface water temperature to variation in the light attenuation coefficient.
The difference in temperature in RefSim and both ExtMinSim, ExtMaxSim is less than 1 ° C.

*ALBM model.* The warmest model in terms of the epilimnion temperature was ALBM model (Fig. 2 (a)). The maximum
water temperature reached in the model is 32.4°C (28.4°C – in observations). However, ALBM demonstrates the lowest

$RMSE_c$ (1.53° C ) and the highest correlation (0.98) with the observed data for the temperature of the water column (except
the surface) during the entire simulation period among all models (Fig. 3 (a), and see SI, Fig. S2, Table S4). It can be noted
visually, that the shape of the thermocline is better reproduced by this model than by other model-participants (Fig. 3 (a)). In
addition, the correlation coefficient of ALBM temperature decreases more slowly with depth than in all other models (Fig. 3
(b)). In contrast, $RMSE_c$ for the surface temperature in ALBM is almost 1.5 larger than in $k - \varepsilon$ models (see SI, Fig. S2 (a)).




The ALBM model successfully reproduces autumnal overturn, although the homogeneous temperature distribution in this model occurs approximately 7-10 days later compared to the measurements. The likely reason for this offset is the excessive accumulation of heat in the mixed layer during summer, which delays the cooling to the temperature of maximum density and subsequent vertical mixing.

5     The ice-cover and ice-off dates are quite close to observations (e.g. 24.12.2011 vs 28.12.2011, see SI, Table S3). Differences between model and observations ice-cover dates by of more than two weeks happen three times.The large errors mainly occur for ice-cover, likely due to the excessive heat accumulated in summer.

***MTCR-1 model.*** In terms of temperature, the Henderson-Sellers-based MTCR-1 model shows contrasting performance compared to ALBM (Fig. 2 (e)). It is generally the coldest model (the temperature reaches a maximum of 24.2 °C in the 10   epilimnion) and results in the deepest mixed layer (5.9 m versus 3.6 m observed). This significant difference in mixed layer depth (see SI, Table S2) is possibly associated with different schemes of convective mixing.

Similar to the FLake model, the surface temperature in MTCR-1 is sensitive to changes in lake depth. The surface temperature DM reduced 4 times compared to the reference model simulation when decreasing the lake depth from 37.5 to 13.32 m (see SI, Table S4).

15     While MTCR-1 and LAKEoneD use similar parameterisation for ice formation and melt, their results are similar only for the ice-off dates. The ice-cover dates greatly vary between these models with differences reaching up to 1 month (winter 2011-2012). Interestingly, LAKEoneD produces ice melt periods in winters 2011-2012 and 2012-2013 (see SI, Table S3) when MTCR-1 does not show it at all (there is no observed ice melt during these winters).

***LAKEoneD and LAKE models.*** Even though LAKE and LAKEoneD use the same turbulence closure, they represent the 20   temperature dynamics in the lake with considerable differences (Fig. 2 (c,d)). In particular, the mean depth of the mixed layer as simulated by LAKE is 1 m deeper compared to LAKEoneD results (4.8 m and 3.9 m, respectively). LAKE demonstrates greater heating of the epilimnion (up to 30 °C) than LAKEoneD (maximum value of 27.3 °C). The homogeneous distribution of temperature during autumn mixing predicted by LAKEoneD reaches the depth of 15 m in most years, as opposed to LAKE (up to 35 m) and the observational data (at least up to 25 m, where the deepest sensor is located). Remarkably, only in these 25   two models periods of unstable ice-cover occur. For example, the ice in LAKE is thin and often comes off during winter (more than 5 times in winter 2011-2012), leading to wind-driven surface layer mixing (see SI, Table S3).

The sensitivity experiments with varying light attenuation coefficient (see SI, Table S4) generally result in a similar response in all models except for FLake. Increasing the light attenuation coefficient (i.e. reducing water transparency) leads to an upward shift of the thermocline position (Fig. 4), and conversely, decreasing the attenuation coefficient causes a downward move of 30   the thermocline. This leads to temparature change at respective depths up to 8 $°C$. Fig. 4 contains only 2 models of different types of turbulent closure because the effect of the varying the light attenuation coefficient is the same in all of them.

It is important to note that all models have a maximum error, $RMSE_c$ for temperature (compared to observations) in the thermocline (Fig. 3 (a)), which agrees with a previous study (Stepanenko et al., 2014). Conversely, the correlation between simulated and observed temperature decreases with depth in all models (Fig. 3 (b)): starting from 0.9-1.0 near the surface to 35   0.1-0.7 close to the bottom.





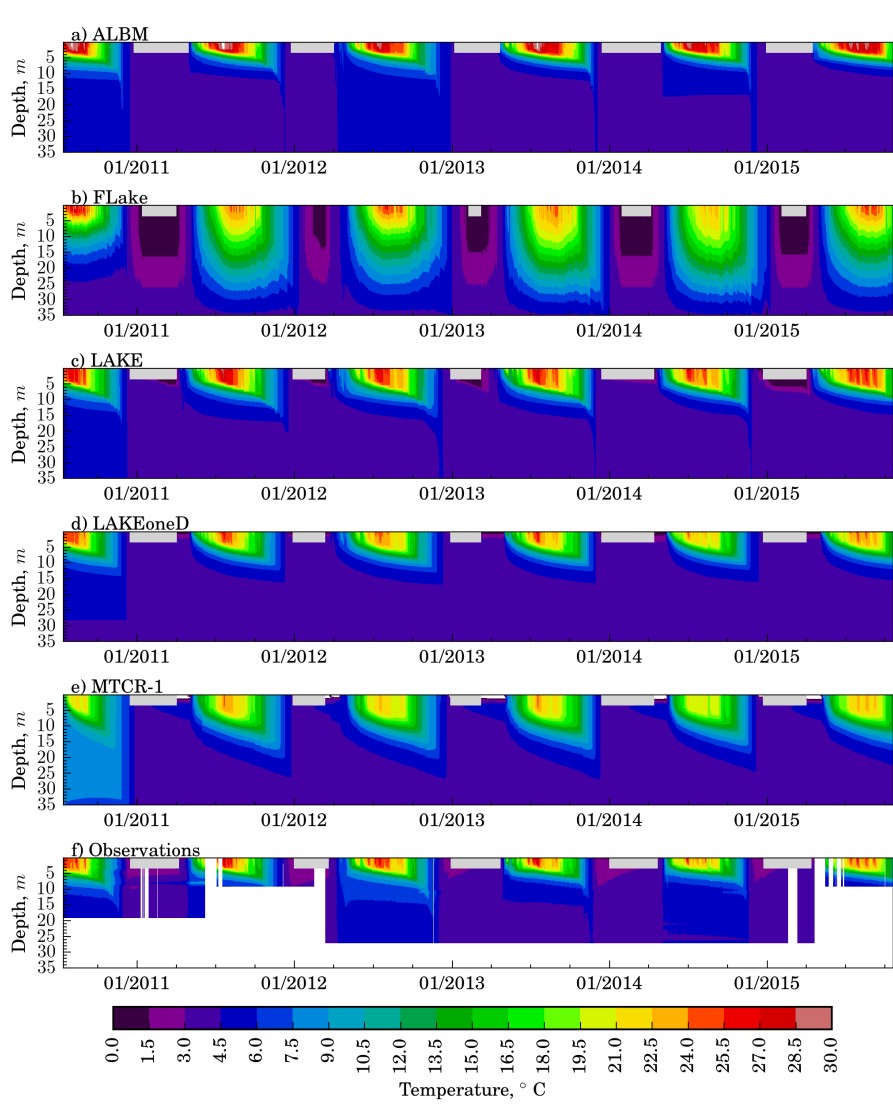

**Figure 2.** Time-depth pattern of temperature in Harp Lake (14.07.2010-19.10.2015), RefSim and observed data. The grey boxes represent duration of the ice-cover period, the white patterns denote the absence of data.





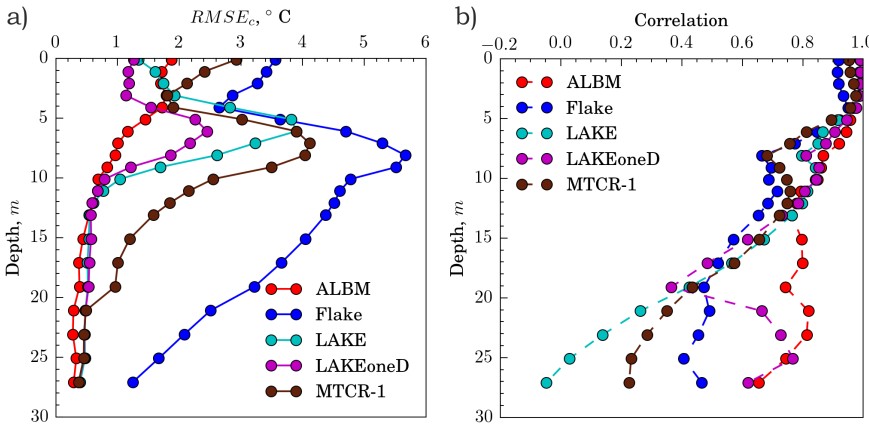

**Figure 3.** (a) Centered root-mean-square error of modeled temperature calculated for each individual depth; (b) the correlation coefficient between the simulated and observed temperature at each depth.

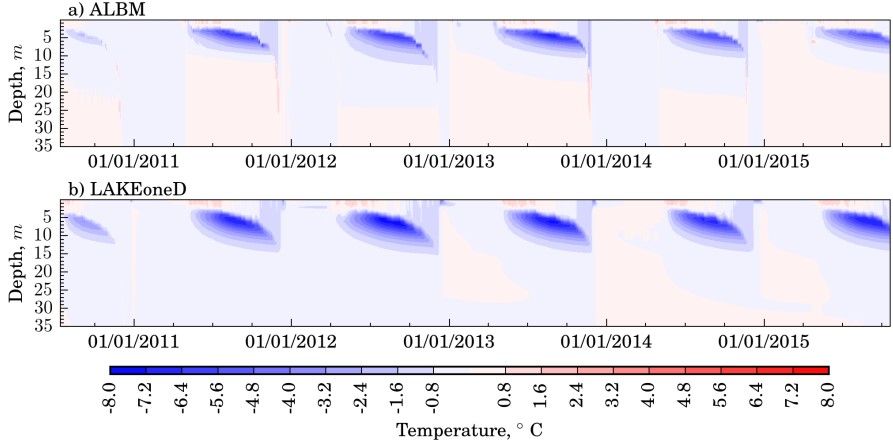

**Figure 4.** Temperature difference between the experiments ExtMaxSim and RefSim in two lake models: (a) ALBM; (b) LAKEoneD.





## 3.2 Eddy diffusivity

Eddy diffusivity (ED), $K$, is the variable controlling the vertical turbulent transport in lake models. This section examines the results of simulated ED values, thereby focusing on discrepancies between models caused by different turbulent closure schemes.

The simulated time-depth distribution of ED is shown in Fig. 5. ED in ALBM,and MTCR-1 do not reproduce spring and autumn whole-water-column mixing as the Henderson-Sellers diffusivity is not valid for unstable stratification. In contrast, $k - \varepsilon$ models correctly predict the deepening of the mixed layer in the ED pattern during summertime towards a complete mixing during autumn.

Even models employing the same type of turbulence parameterisation differ in simulated eddy diffusivities in the thermocline
by orders of magnitude. This is caused by the different background values of ED added to those estimated by turbulence closures and mimicking mixing mechanisms not directly resolved in 1D models such as internal waves breaking (e.g. Hondzo and Stefan (1993)). Apparently, the background diffusivity is switched off in MTCR-1 model during ice-cover.

The increase of ED near the bottom during the stratified period in the ALBM can be explained by the change from a stable stratification in the thermocline to a quasi-neutral stratification beneath.

A peculiar feature of LAKE is that it features high-intensity turbulence during much of the ice-cover period in the lower part of the water column that is apparently related to residual flow after the autumn overturn. In contrast, in LAKEoneD, turbulence dissipates shortly after the ice-cover. Both of these models demonstrate a subsurface region of high ED values in winter, which is not reproduced by other models. Production of turbulent kinetic energy (TKE) below the ice may be caused by both momentum transfer through partial ice-cover (implemented in LAKE) and shortwave radiation penetration causing
under-ice convection. The conspicuous difference between LAKE and LAKEoneD results is that springtime deep mixing does not develop in LAKEoneD. This may be linked to different constants and stability functions used in its $k - \varepsilon$ closures, or different buoyancy and momentum fluxes at the lake-atmosphere interface during this season.

It is worthy to note that the significant difference between the modeled and observed dates of ice-cover and ice-off (see SI, Table S3) leads to errors in the simulation of ED as water opens and interacts with the atmosphere.

The main features of ED distribution over seasons described in this section hold in experiment LDSim with different lake depth value (13.32 m).



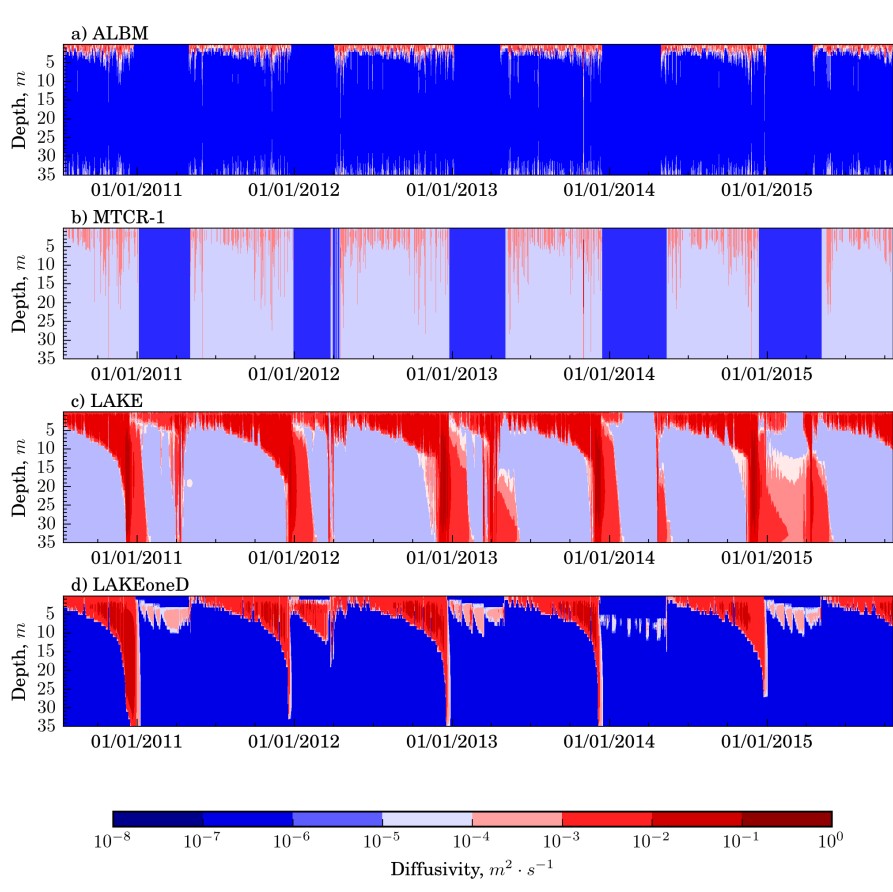

**Figure 5.** Time-depth distribution of the ED in Harp Lake (14.07.2010-19.10.2015), RefSim model experiment.



### 3.3 Effects of eddy diffusivity on the vertical distribution of a passive tracer

The vertical distribution of ED in lake models is one of the major drivers of vertical transport of dissolved gases and their emission to the atmosphere. To isolate the effect of the ED parameterisation in the different models, we solve the diffusion equation for a passive tracer concentration $C$:

$$\frac{\partial C}{\partial t} = \frac{\partial}{\partial z} K \frac{\partial C}{\partial z} . \tag{1}$$

The emission rate of the passive tracer at the bottom is set constant:

$$K \left. \frac{\partial C}{\partial z} \right|_{z=h} = F_{bot} , \; F_{bot} = const > 0 , \tag{2}$$

while at the surface the flux is assumed to be large enough to make the surface concentration negligible compared to deep-water concentration, so that the boundary condition here becomes:

$$C|_{z=0} = 0 . \tag{3}$$

Setting zero concentration at the surface is justified by the fact, that surface concentration of gases of interest ($CO_2$, $CH_4$) is typically orders of magnitude less than bottom concentration. The total (molecular plus turbulent) diffusivity $K$ is taken from different models, so that the solution $C$ of Eq. (1)-(3) is individual for each lake model.

Calculation of tracer diffusion was carried out using simulated values $K(z,t)$ from the two Henderson-Sellers-based models – ALBM, MTCR-1, and two $k-\varepsilon$ models – LAKE and LAKEoneD, covering the period of 5 years. In order to take into account vertical mixing under unstable stratification performed in ALBM and MTCR-1 and not expressed by Henderson-Sellers diffusivity, a value $K = 10^5$ m$^2$/s was applied for the convective layer in such cases.

The results of the numerical tracer experiment are presented in Fig. 6. All models demonstrate seasonal variability of the passive tracer concentration, including the near-bottom accumulation during stratified periods. Three models (ALBM, MTCR-1, and LAKE) perform the full overturn of the water column during autumn and spring, leading to the emission of almost the whole tracer storage to the atmosphere and forming a homogeneous concentration profile. In LAKEoneD, the vertical mixing does not reach the lake bottom during spring and autumn in certain years (e.g. 2011 and 2014). Hence, not all tracer amount is removed from the water column during overturn periods (see SI, Fig. S3 (b)) in this model. As a result, not only the concentration of the tracer near the bottom, but also the total tracer amount in water column increases in LAKEoneD throughout the simulation period. However this effect is a result of the assumption of a bathtub shaped lake bathymetry and would be less when including the depth dependence of lake volume.

The MTCR-1 model demonstrates short overturn periods similar to those of LAKEoneD, but the mixing reaches deeper. Hence, the passive tracer does not accumulate over time. Furthermore, ALBM exhibits a higher near-bottom concentration than MTCR-1 because the ED simulated by ALBM is one order of magnitude lower compared to MTCR-1 at 30-35 m depth (see Fig. 6).



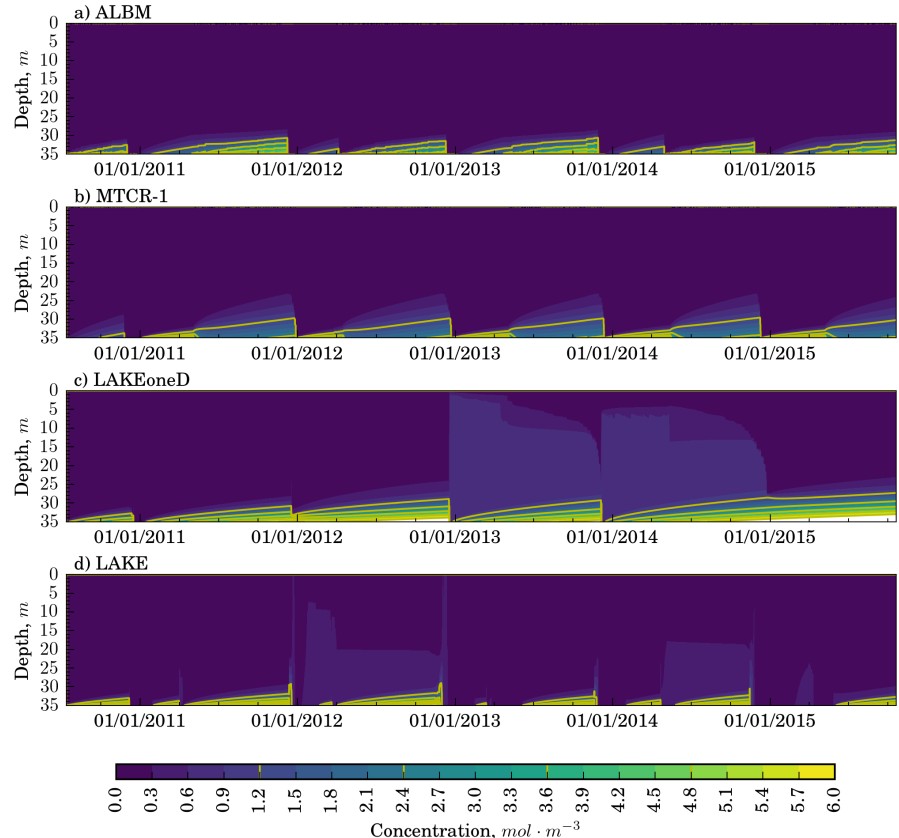

**Figure 6.** Vertical distribution of passive tracer concentration in Harp Lake simulations, according to different models

To assess the effect of different eddy diffusivities on the surface tracer flux, we use the ratio of the surface flux, $F_{surf}$, to the bottom flux, $F_{bot}$, where the former is given as:

$$F_{surf} = K_{surf} \cdot \left( \frac{C_w - C_{eq}}{\Delta z} \right), \tag{4}$$

with $C_{eq} = 0$ the surface concentration, $C_w$ the concentration at the first model output level below surface (0.3 m), both in

5  [mol m$^{-3}$], $\Delta z = 0.3$ m, and $K_{surf}$ is surface diffusivity coefficient.

The Henderson-Sellers-based models produce short periods with high fluxes, primarily during autumnal mixing (see SI, Fig. S3 (a)) and the drops of the integral concentration in the water column at the same time (see SI, Fig. S3 (b)). The flux spikes are several orders of magnitude higher compared to $k - \varepsilon$ models. A reasonable explanation is that instantaneous convective mixing implemented in ALBM, MTCR-1 models leads to almost the whole tracer content inside the convective

10 layer to be released to the atmosphere during a single convective event. All models demonstrate much weaker tracer fluxes during spring overturn compared to those resulting from autumnal convection.



Lake models with $k-\varepsilon$ closure produce substantial tracer fluxes during the stratified period in particular years (e.g. 2013 and 2014). The likely reason for this effect is rapid changes in mixed-layer depth, due to sudden wind forcing, entraining higher concentrations from the hypolimnion into the mixed layer.

The tracer diffusion experiment was also conducted at the reduced depth of 13.32 m (see SI, Fig. S3 (c,d)). Due to the lower depth, seasonal vertical mixing now extends over the whole water column in all models, providing more effective release of the substance. ALBM and MTCR-1 demonstrate episodic high fluxes during summer along with $k-\varepsilon$ models. The spring mixing in LAKEoneD now reaches the bottom in some years, hence the integral tracer amount evolves here in a similar way compared to all other models. The time-averaged tracer emission in LAKEoneD is thus the most sensitive to depth reduction. The mean surface flux in this model increases up to approximately 54 %, while for ALBM, MTCR-1, LAKE – this increase is 4 %, 19 %, and 8 %, respectively. This can be primarily explained by the contribution of the increased fluxes during summer stratification and of the autumn mixing now reaching the bottom.



### 3.4 Harp Lake biogeochemistry

#### 3.4.1 Oxygen

The evolution of observed vertical distribution of $O_2$ concentration in Harp Lake is shown in Fig. 7 c excluding winter periods and data with higher sampling frequency (1 h) collected at 1, 18 m depth (Fig. 8). During spring (end of April - beginning of

May) after ice-off, convective mixing occurs, leading to a homogenized $O_2$ profile. The similar process happens in autumn. During summer, $O_2$ depletes in the hypolimnion due to its consumption by organic matter degradation in sediments and deep water layers. The main source of $O_2$ is located in the photic layer, where it forms as a result of photosynthesis. Below the mixed layer (at 5-10 m depth), there is a maximum of $O_2$ concentration due to weak turbulence, so that the produced $O_2$ is not transported to the surface and emitted to the atmosphere. In particular, the $O_2$ maximum is remarkable during the summer

of 2012: ($C_{O_{2 max}}$ increases up to 15 mg l$^{-1}$) at 5 m depth. During winter, the $O_2$ concentration is usually high, indicating high saturation and low $O_2$ consumption rate from organic matter degradation in the lake. In summer surface oxygen content decreases following reduction of temperature-controlled solubility. The measurements thus indicate common trends observed in most deep oligotrophic lakes at mid-latitudes (Mitchell and Prepas, 1990).

Both models depict a reasonable representation of $O_2$ dynamics at different depths (Fig. 7 (a, b)). ALBM includes the

$O_2$ model described in Tan et al. (2017), whereas LAKE employs the representation of $O_2$ sinks and source from (Stefan and Fang, 1994). Even though they use different oxygen and turbulent parameterisations, both models show similar spatio-temporal pattern. In particular, they reproduce the maximum of $O_2$ concentration below the mixed layer, its quasi-linear in temporal decrease in the hypolimnion (18 m) during stratified periods, and the effect of mixing events during spring and autumn, marked by rapid changes of $O_2$ concentration at 18 m depth. However, the $O_2$ concentration as computed by LAKE

is on average 1-2 mg l$^{-1}$ higher than the observed values. A plausible explanation for this positive bias is that photosynthesis in LAKE yields excessive production of $O_2$ (parameters of biogeochemical parameterisations in this model have not been calibrated in the study). Another deficiency of LAKE simulations is that during ice-cover near-surface $O_2$ decays linearly, and then exhibits significant peaks at thinning of ice-cover and ice-off, whereas in ALBM and measurements, oxygen content holds almost constant during these periods.

The time series of $O_2$ concentration from ALBM and LAKE correlate reasonably to observed data, with linear correlation coefficient ranging between 0.5-0.6 at 1 m, 0.6-0.8 at 18 m depth, and with RMSE$_c$ 1.27-1.56 mg l$^{-1}$ at 1 m, 1.19-1.52 at 18 m depth (see SI, Fig. S4). For both depths, ALBM demonstrates higher correlation with observations. Both models correlate better with observations at large depth. Decrease of the temperature in the ExtMaxSim simulation (Fig. 4 due to the upward shift of thermocline leads to the reduction of $O_2$ concentration at the respective depths.

Overall, the oxygen content is highly dependent on physical controls which are temperature, radiation, ED and ice-cover. Given that these factors are reasonably reproduced by ALBM and LAKE, $O_2$ spatiotemporal pattern is captured as well.



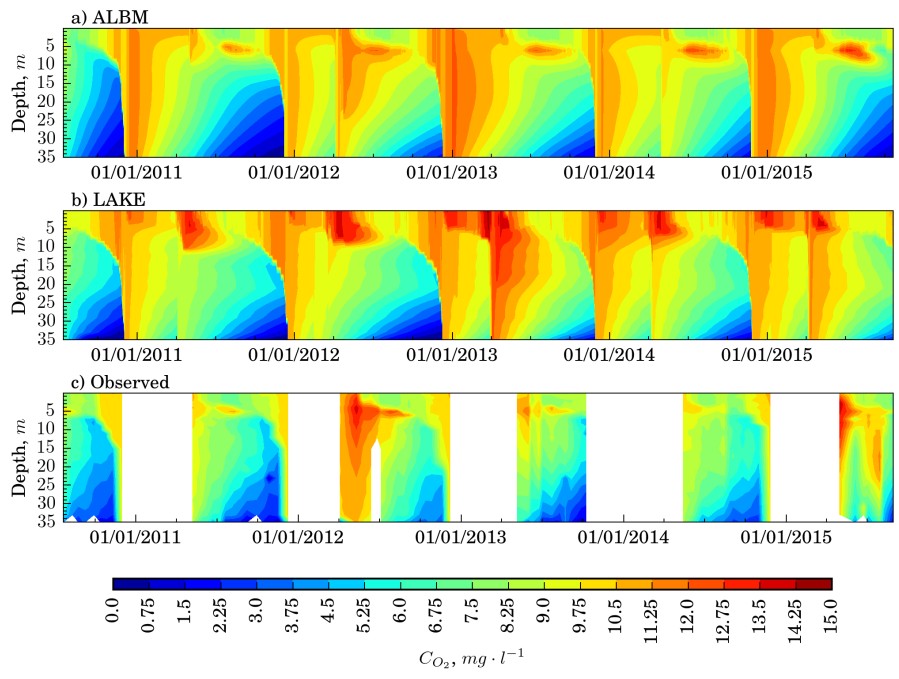

**Figure 7.** Time-depth profiles of $O_2$ in Harp Lake (14.07.2010-19.10.2015), model experiment including sediments and morphometry (Gas-Sim). White patterns in (c) indicate absence of data.

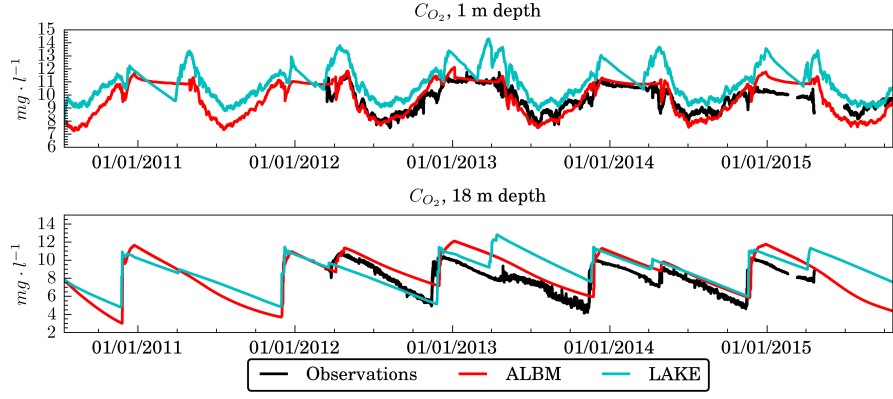

**Figure 8.** Time series of $O_2$ concentration at depths 1 m and 18 m, observed and simulated



### 3.4.2 Carbon Dioxide

The vertical distribution of $CO_2$ concentration simulated by ALBM and LAKE are shown in Fig. 9. ALBM simulates higher concentrations of $CO_2$ compared to LAKE. This is likely because ALBM considers dissolved inorganic carbon (DIC) inflow from the catchment and explicitly treats DOC and particulate organic carbon (POC) dynamics (Tan et al., 2017). Parameter

calibration applied in ALBM brought $CO_2$ content closer to measurements in this model. Correlation with the observational data at depth of 0.39 m is relatively small for the both models (0.3) and $RMSE_c$ is 2.7, 4.5 mg $l^{-1}$ for LAKE and ALBM, respectively.

Both models reproduce the seasonality of the $CO_2$ concentrations: during summer $CO_2$ accumulates in the hypolimnion, while $CO_2$ is consumed in the mixed layer due to photosynthesis. For the same reason, the depth of the $O_2$ maximum (Fig. 7

(c)) is consistent with the metalimnetic $CO_2$ minimum.

Vertical turbulent diffusion (Sec. 3.3) greatly affects $CO_2$ patterns in models. In particular, the mixing periods clearly seen in Fig. 9 coincide with those in Fig. 6. Accumulation of $CO_2$ below mixed layer during stratified periods is clearly demonstrated by both models. However, the vertical $CO_2$ distribution in the thermocline is very different, i.e. in LAKE, concentration attains its maximum near the bottom, whereas in ALBM the maximum is located at 15-20 m depth. This can be related to the fact that

in ALBM, DOC and DIC from surface water and ground water can be injected in the middle of the water column.

In contrast to oxygen, measured $CO_2$ content is much less correlated to simulated. The wintertime increase of surface concentration seen in calculated series is hardly observable in measured data. In summer, simulated $CO_2$ series are smooth, whereas there are significant fluctuations in empirical data. There is some correspondence between seasonal-mixing-caused peaks in modeled concentration and measured one, though with time lags. All this leads to conclusion, that, compared to

oxygen, carbon dioxide is much more controlled by biogeochemical processes misrepresented in lake models, than by physical factors.





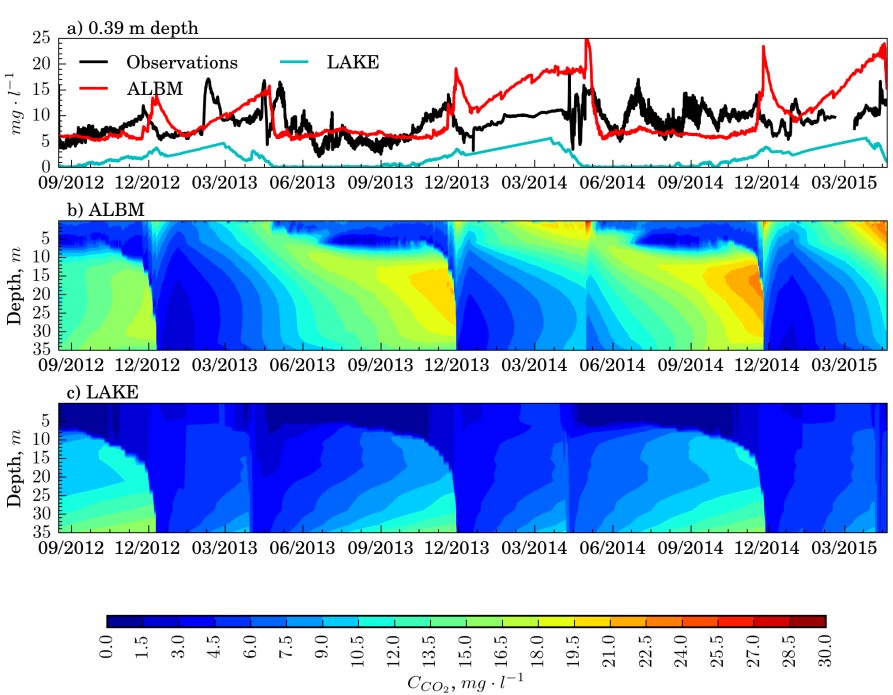

**Figure 9.** Time-depth profiles of $CO_2$ concentration in Harp Lake (17.08.2012-20.04.2015), experiment including modules of sediments and morphometry (GasSim)





## 4 Conclusions

A lake model intercomparison study focusing on biogeochemical processes (i.e., dissolved gas concentration) is performed, making a step forward in the LakeMIP exercises, which have been previously focused on thermodynamic modeling only. To evaluate model performance with respect to biogeochemical processes we had to first evaluate their performance in hy-
dro/thermodynamic simulations and variations due to uncertainty in major driving parameters, such as temperature and turbulent diffusivity.

The participating lake models (ALBM, FLake, LAKE, LAKEoneD, MTCR-1) were evaluated regarding their ability to reproduce the thermodynamic regime of Harp lake. All models capture the seasonal variability of the temperature profile in the deep boreal dimictic lake. A substantial discrepancy was found between models in representation of thermocline evolution –
consistent with previous studies – and the correlation coefficient for calculated and observed temperature was found to decrease from surface to bottom (from 0.9-1 to 0.1-0.6). The sensitivity of the simulation results to the light attenuation coefficient and the lake depth was assessed, given that in global applications, such as numerical weather prediction, they are uncertain external parameters.

All considered lake models cannot precisely reproduce the dates of ice-cover and off, similar to the results from a previous
simulation study for the same lake which used other lake models and an earlier time period (Yao et al., 2014). The difference between the model results and the observations exceeds 2 weeks in particular years. The unrealistic representation of ice formation and ice melt dates leads to untimely changes in eddy diffusivity (ED) within the water column, which in turn strongly affects the modeled distribution of gases and their emission to the atmosphere. In a recent study, (Tan et al., 2018) one possible direction to improve the simulation of ice-cover is shown. They demonstrated that including the conversion of snow
to white or slush ice when the weight of ice and snow exceeds the buoyancy of the ice cover, can significantly improve the ice simulation results.

To study the effect of turbulent transport on the vertical distribution of gases, we conducted passive tracer simulations using simulated ED from different models.

This experiment reveals that all lake models demonstrate more intense mixed layer deepening during autumn than during
spring, with spring overturn in some cases not even reaching the lake bottom (e.g. in MTCR-1, and LAKEoneD).

Moreover, less intensive mixing during spring and autumn leads to an accumulation of passive tracer concentrations in deep water at a multiyear timescale.

In addition, the representation of an equivalent lake depth in 1-dimensional models is found to be a crucial parameter regulating both passive tracer concentration and its flux to the atmosphere: when reducing the depth of a lake approximately
by 50 %, the average tracer flux increased by 4 %-54 %.

The lake models with most comprehensive biogeochemical modules such as ALBM, LAKE demonstrated reasonable reproduction of the $O_2$ concentration profile and its seasonal evolution. For instance, both models predict an oxygen concentration maximum and corresponding $CO_2$ minimum below the mixed layer in summer, elevated surface concentration and linear decay of deep-water content under ice-cover, vertical homogenization during spring and autumn overturns.



The main difference between the models in the $CO_2$ simulation is that the ALBM model involves additional source of DIC from the catchment. Due to this, the average $CO_2$ concentration in ALBM is higher than in LAKE and closer to measurements. Results from both models are weakly correlated to measurements of surface $CO_2$ concentration. This leads to conclusion, that, compared to oxygen, carbon dioxide is much more controlled by biogeochemical processes misrepresented in lake models, than by physical factors.

As a result of this study, we emphasize the following issues to be addressed by the lake modeling community in respect to dissolved gases simulation:

- simplified lake modeling approaches especially employing parameterized temperature profile and representing reasonably the surface temperature may fail in calculating vertical temperature distribution with potentially significant effect on biogeochemical processes, if the respective module is coupled to thermodynamic compartment;

- for deep lakes, models differ in the depth of autumn and especially spring overturn; if this depth reaches bottom, the gas content originated from sediments and accumulated in the hypolimnion during the preceding stratified period is almost completely released to the atmosphere, otherwise these gases may continuously accumulate in deep water at a multi-year timescale; however, more research is needed to establish the effect of lake morphometry on these processes;

- oxygen content in both surface and deep water is efficiently controlled by physical and biogeochemical factors, successfully simulated by lake models, whereas measured surface $CO_2$ exhibits significant temporal variability not captured by the models, calling for more research in biogeochemical mechanisms and parameterisations of carbon dioxide dynamics.

## 5  Code and data availability

| Model/Data | Code and data availability |
| --- | --- |
| ALBM | Available upon request to Zeli Tan, e-mail: tanzeli1982@gmail.com |
| MTCR-1 | Available upon request to Bruna Arcie Polli, e-mail: brunapolli@gmail.com |
| FLake | Publicly available at the website: http://www.flake.igb-berlin.de/ |
| LAKEoneD | Available upon request to Klaus Jöhnk, e-mail: science@limnophysics.de |
| LAKE | Publicly available at the website: http://tesla.parallel.ru/Viktor/LAKE/wikis/LAKE-model |
| Observational data | Available upon request to Huaxia Yao, e-mail: huaxia.yao@ontario.ca |

## 6  Author contribution

V.S., W.T., K.J., Z.T., H.Y. conceptually designed the experiments. S.G., B.A.P., T.B., W.T., K.J., Z.T., Q.Z. carried them out and provided the results of the simulations. J.A.R. and H.Y. provided observational data. S.G. made a formal analysis and



combined all the data, visualized them under the supervision of V.S. S.G. prepared the manuscript under the supervision of A.L., V.S. with significant contributions from all co-authors. The authors declare that they have no conflict of interest.

*Acknowledgements.* Simulations performed with LAKE model have been supported by Russian Science Foundation (grant 17-17-01210). The research is partially supported by a USGS project (G17AC00276) to Q.Z. Z.T. is supported by the U.S. DOE's Earth System Modeling

5     program through the Energy Exascale Earth System Model (E3SM) project. The supercomputing resource for ALBM model simulations is provided by Rosen Center for Advanced Computing at Purdue University. W.T. is supported by an ETH Zurich postdoctoral fellowship (Fel-45 15-1). The Uniscientia Foundation and the ETH Zurich Foundation are acknowledged for their support to this research. Thanks are given to DESC staff (e.g. R. Ingram, C. McConnell, T. Field) for contributing to data collection and to the Ontario Ministry of the Environment and Climate Change for funding the traditional and high-frequency lake monitoring activities.Simulations of MTCR-1 were supported by the

10    brazilian funding agencies CNPQ (grant number: 308758/2017-0) and CAPES .





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
