# Peer review of "Multimodel simulation of vertical gas transfer in a temperate lake"

_Hydrology and Earth System Sciences, 2019_

## Referee Comment (RC1) · Anonymous Referee #1 · 2 Aug 2019

The study continues a series of previously published lake model intercomparison efforts by extending those on modeling of dissolved oxygen and carbon dioxide in lakes. Five models are used in the comparison, whereas only two of them simulate the oxygen and $CO_2$ regimes directly. Three others are confined to modeling of the thermal stratification and ice regime, considered here apparently as "potential candidates" for their extension on biogeochemical processes. As such, the approach is legitime, since the correct simulation of the seasonal thermal stratification and ice duration is the key prerequisite for adequate modeling of the dissolved gases transport between their major sources and sinks at the surface and the bottom of a lake.

Concerns arise however about the way one of the models—FLake—was treated in the study. The authors correctly state in the description of the model experiments that

[Figure]

FLake "stands aside from the other 1-D models due to the ... bulk-structure which employs the concept of <temperature profile> self-similarity...". This high level of parameterization ensures computational efficiency of the model, which was primarily designed for prediction of surface temperatures in global/regional climate models and numerical weather prediction (NWP). On the other hand, the model parameterizations put some constraints on the model application to real lakes. One crucially important feature of FLake is that the model equations are derived in the assumption of preserving the heat capacity or *volume of the lake*. In this regard, the "baseline" configuration applied in this study with the maximum lake depth as model input is inappropriate for FLake and would produce a priori incorrect results. Another issue of the parameterized model refers to correct choice of the few "shape factor" parameters determining the stratification pattern. The latter is not resolved in FLake numerically, but parameterized via few "shape factor"-constants related to the spatial integrals over the stratified layer. Several recent publications, including co-authorship of one of the authors of the present study, (Shatwell et al., 2016; Kirillin et al., 2017; Shatwell et al., 2019; Su et al., 2019) discussed the appropriate choice of the shape factor constants in FLake and have demonstrated that the set of constants used in the NWP-version of the model should be amended if the vertical thermal structure is in question apart from the lake surface temperatures. In particular, the unrealistically weak deep stratification and the corresponding high depth of the surface mixed layer, as reported in the present study, are the results of applying the NWP-constants together with the maximum lake depth as the model lake depth.

Hence, the FLake-outcomes discussed here are useless or even misleading for the potential FLake-users.

I see two ways of possible modification of the study: (i) excluding FLake from the study completely, confining the set of tested models to four models; (ii) re-designing the FLake-experiments in the corrected way with corresponding changes in the results/discussion.

Regarding the biogeochemical part, the study would benefit from an extension of the discussion on possible ways of improving the representation of biogeochemistry, in particular, the deep oxygen (chlorophyll) maximum in oligotrophic lakes, and the vertical distribution of carbon dioxide across the water column.

Note also that the temperature profile within the ice cover in FLake is not assumed to be linear, but parameterized via a time-varying shape-function with a linear asymptotic.

**References**

Kirillin, G., Wen, L., and Shatwell, T.: Seasonal thermal regime and climatic trends in lakes of the Tibetan highlands, Hydrology and Earth System Sciences, 21, 1895–1909, 2017.

Shatwell, T., Adrian, R., and Kirillin, G.: Planktonic events may cause polymictic-dimictic regime shifts in temperate lakes, Scientific reports, 6, 24 361, 2016.

Shatwell, T., Thiery, W., and Kirillin, G.: Future projections of temperature and mixing regime of European temperate lakes, Hydrology and Earth System Sciences, 23, 1533–1551, 2019.

Su, D., Hu, X., Wen, L., Lyu, S., Gao, X., Zhao, L., Li, Z., Du, J., and Kirillin, G.: Numerical study on the response of the largest lake in China to climate change, Hydrology and Earth System Sciences, 23, 2093–2109, 2019.

---

## Referee Comment (RC2) · Anonymous Referee #2 · 23 Aug 2019

The present study is a lake model intercomparison exercise conducted in a temperate lake in Canada with a focus on biogeochemical processes in lakes and more precisely carbon dioxide and dissolved oxygen modeling. The study first focuses on the simulation of thermal stratification and ice cover, and then vertical diffusion of gases which are key elements for the vertical transport of greenhouse gases in lakes. Although five models are involved in the intercomparison, only two of them, of higher complexity, are able to model carbon dioxide and oxygen concentrations evolutions. The subject is of real interest to study in details CO2 and O2 dynamics in lakes.

General comments

It is stated that the study is a continuity of LakeMIP exercises accounting for biogeochemical processes comparisons. Although the first part of the paper is dedicated to

thermal stratification and ice cover study, and involves the five models, the fact that only two of them have the possibility to simulate O2 and CO2 dynamics, indicate that this is not a real LakeMIP type exercise, and therefore constitutes a limitation in my view to be considered as a true intercomparison model experiment for vertical gaz transfer.

In the first LakeMIP exercise (Stepanenko et al., 2010) the sensitivity of lake depth has been studied and the experiment setup accounted for simulations with maximum depth, local depth and average depth. And this is crucial especially (maybe only) for FLake model as it was demonstrated that an average lake depth was necessary for FLake simulation in order to be conservative in terms of energy. In the current setup, the maximum lake depth is used for all models and this is contradictory with a correct use of FLake which should be run using the LDSim configuration. It could be interesting to compare in the same graph RefSim and LDSim at least for FLake.

As a consequence, the sensitivity test on light extinction for FLake is not relevant since the thermal profile cannot be well simulated, and therefore should be conducted with a depth of 13.32m.

Specific comments

Looking at the bathymetry indicates strong gradients from the shoreline to the point of maximum depth. There are probably 3d-circulations that take place when dense waters flow along the bottom slopes. And these circulations are not accounted for in the 1d simulations. What is in your view the potential impact on these circulations on the thermal stratification and ice cover? And on modeled vertical transport? It would be interesting to add a discussion on that particular point.

The abstract mentions the need to improve biogeochemical processes in lake models to enhance weather prediction and climate projection capabilities. I'm not convinced improving biogeochemical processes will improve weather prediction. What is crucial in weather prediction or climate modeling is to simulate a correct surface temperature and fluxes because these are the variables that will be used in the coupling to the

atmosphere. That's true that in climate simulations the knowledge of carbon dioxide or methane emissions are of high interest, however to my knowledge only the LAKE model offers the capability to simulate CO2, O2 and CH4 dynamics and be coupled to a climate model. Please add a discussion on that point to also highlight the difficulty to increase the complexity of lake models and ensure a correct coupling to an atmospheric climate model.

Ice cover is a key variable for vertical transfer of gases. It has been shown that freeze-up or brake-up presented delays of several weeks potentially. Don't you think more effort should be put on the representation of ice and snow over ice in lake models, especially when working in NWP and/or climate contexts?

It would also be of interest to have a comparison of surface temperatures observed at 10cm to the model simulations to be sure that the daily cycle of temperature is well reproduced. This is a key feature for any further coupling to an atmospheric model. Could you add such a graph in the revised manuscript?

A comparison of methane profiles for ALBM and LAKE would also be interesting (climate change context, . . .) even if no observation is available.

Technical comments

Page 4 line 12: change – by of

Page 6 line 5: summarized

---

## Author Comment (AC2) · 5 Nov 2019

**General comments**

1. (1) It is stated that the study is a continuity of LakeMIP exercises accounting for biogeochemical processes comparisons. Although the first part of the paper is dedicated to thermal stratification and ice cover study, and involves the five models, the fact that only two of them have the possibility to simulate O2 and CO2 dynamics, indicate that this is not a real LakeMIP type exercise, and therefore constitutes a limitation in my view to be considered as a true intercomparison model experiment for vertical gaz transfer.

(2) Dissolved gas transport throughout the water column is mainly carried out by turbulence. Despite the fact that only two of the models have an ability directly to reproduce the concentration the gases (oxygen and carbon dioxide) our numerical experiments with a passive tracer (which can be seen as a "prototype" of gas or other constituent) involving four models out of five showed the potential simulation of gases governed by seasonal stratification and ice cover. We can consider this experiment as the inter-comparison model experiment in respect to primary physical controls of gas dynamics. We consider the intercomparison experiment involving full biogeochemical models as a next step of this study.

(3) No comments/corrections are added to the manuscript.

2. (1) In the first LakeMIP exercise (Stepanenko et al., 2010) the sensitivity of lake depth has been studied and the experiment setup accounted for simulations with max-imum depth, local depth and average depth. And this is crucial especially (maybe only) for FLake model as it was demonstrated that an average lake depth was necessary for FLake simulation in order to be conservative in terms of energy. In the current setup, the maximum lake depth is used for all models and this is contradictory with a correct use of FLake which should be run using the LDSim configuration. It could be interesting to compare in the same graph RefSim and LDSim at least for FLake.

(2) The main focus of our study is the vertical diffusion which is the main driver for the gas distribution. Of major interest are such gases as carbon dioxide and methane which are primarily produced in lake sediments. Setting the maximum lake depth be-comes crucial in terms of the total vertical distance required for transport of bottom-originated gases to the atmosphere. For this reason, we conduct the baseline exper-iment with the maximum depth. We added new figures to the manuscript with results from the LDSim experiment – FLake showed the largest sensitivity to the variation of the lake depth among the other models: e.g. $RMSE_c$ for surface temperature as well as for the whole temperature profile reduced almost twofold (see Table S4, Fig. S3, S4). This suggests potential limitation of FLake for correctly simulating gas dynamics in deep lakes (given the biogeochemical block is added to the model).

(3) Page 10 line 24: Remarkably, FLake demonstrates the largest sensitivity among all models to the lake depth – $RMSE_c$ is reduced approximately twofold (from 2.5 ° C to 1.28 ° C) (see SI, Table S4, Fig. S3, S4) when using a mean depth of 13.32 m instead of the maximal depth 37.5 m. The thickness of mixed layer depth reduces up to 4.5 m and have a better agreement with the observations in comparison with RefSim experiment. We added the mean depth of the mixed layer for all models in the LDSim experiment to the Table S2 in SI. We also put two additional figures for LDSim simulation to SI: Fig. S3, S4.

3. (1) As a consequence, the sensitivity test on light extinction for FLake is not relevant since the thermal profile cannot be well simulated, and therefore should be conducted with a depth of 13.32 m.

(2) We redid all FLake simulations using calibrated shape factor, and temperature profile is now much better reproduced compared to results presented in the first version of the manuscript. Correspondingly, the sensitivity test on light extinction coefficient with FLake demonstrates now result similar to other models (see new Fig. 4).

(3) Page 11 line 32: This leads to temperature change at respective depths up to 4 $°C$ in FLake model and 8 $°C$ in ALBM and LAKEoneD. Fig. 4 contains only 3 models with different types of turbulent closure because the effect of varying the light attenuation coefficient is very similar in models on the same type (e.g. k-$\epsilon$ models LAKEoneD and LAKE).

We also added the result of ExtMaxSim simulation for FLake in Fig. 4 in the manuscript.

**Specific comments**

4. (1) Looking at the bathymetry indicates strong gradients from the shoreline to the point of maximum depth. There are probably 3d-circulations that take place when dense waters flow along the bottom slopes. And these circulations are not accounted for in the 1d simulations. What is in your view the potential impact on these circulations

on the thermal stratification and ice cover? And on modeled vertical transport? It would be interesting to add a discussion on that particular point.

(2) 1D models have certain limitations when it comes to such a 3-dimensional system as lake. They do not capture all the lake mixing mechanisms such as density driven currents, which can be important under certain conditions. In general, these currents are very slow (only a few cm s$^{-1}$) (Bengtsson, 2012) and are not important in terms of the turbulence production. The strong contribution of these flows in the vertical exchange of deep and near-surface water was found for deep lakes (> 100 m) with an extended literal zone, e.g. Lake Geneva and Lake Van (Fer et al., 2001, Fer et al. 2002, Kaden et al., 2010). For Harp Lake, in particular, this process may not be important during summer due to a smaller depth and a limited area of shallow part. In winter, a large-scale convective circulation (up to 3-5 cm s$^{-1}$) due to heat exchange at the sediment-water interface may develop under ice (Kirillin et al., 2015). However, the importance of this circulation in terms of gas transfer have not been studied yet. At the moment, this type of the lake circulation is a topic of the ongoing research and it is not included into 1D models. Modeling such 3-d phenomena requires an application of 2D or 3D models.

(3) Added to Page 24 line 22 : - 1D models have certain limitations when applying to such a 3-dimensional system as lake. They do not capture all the lake mixing mechanisms such as density driven currents, which can be important under certain conditions (Samolyubov,1999). It was found that for deep lakes with extended literal zone, such as Lake Geneva (Fer et al., 2001, 2002) and Lake Van (Kaden et al., 2010), these flows can be significant in terms of the vertical heat and gas transfer. In particular, for Harp Lake it may not be important due to a smaller depth and a not large shallow area. In winter, a large-scale convective circulation (up to 3-5 cm s$^{-1}$) due to heat exchange at the sediment-water interface may develop under ice (Kirillin et al., 2015). However, the importance of this circulation in terms of gas transfer have not been studied yet. So far, to the best of our knowledge, these currents are not parameterized in 1D models.

5. (1) The abstract mentions the need to improve biogeochemical processes in lake models to enhance weather prediction and climate projection capabilities. I'm not convinced improving biogeochemical processes will improve weather prediction. What is crucial in weather prediction or climate modeling is to simulate a correct surface temperature and fluxes because these are the variables that will be used in the coupling to the atmosphere. That's true that in climate simulations the knowledge of carbon dioxide or methane emissions are of high interest, however to my knowledge only the LAKE model offers the capability to simulate CO2, O2 and CH4 dynamics and be coupled to a climate model. Please add a discussion on that point to also highlight the difficulty to increase the complexity of lake models and ensure a correct coupling to an atmospheric climate model.

(2)-(3) As in this study we do not focus on methane, due to absence of respective measurement data, and discuss models performance for $CO_2$ content, we added a comment on issues which will arise when implementing lacustrine $CO_2$ emissions in climate models (Page 21 line 29): The important role of catchment processes in building up the DIC levels in lakes introduces an extra difficulty for implementing lacustrine $CO_2$ emissions in the Earth system models. This is caused by the necessity to provide regional or global data on lake's catchments geometric, physical and biogeochemical properties, which are not currently available. In relation to this, we can also note a faster progress on a roadmap to introduce lake $CH_4$ dynamics in climate models, where simulations passed from site-levels studies to regional estimates (Tan et al., 2015), whereas $CO_2$ modeling is currently confined to individual lakes (Stepanenko et al., 2016; Tan et al., 2017; Kiuru et al., 2018).

6. (1) Ice cover is a key variable for vertical transfer of gases. It has been shown that freezeup or brake-up presented delays of several weeks potentially. Don't you think more effort should be put on the representation of ice and snow over ice in lake models, especially when working in NWP and/or climate contexts?

(2)-(3) We agree and added the following comment at Page 23 line 18: This implies

more efforts of a community should be spent on elaborating ice-snow schemes in lake models, where currently vertical homogeneity and temporal invariance of optical and thermodynamic properties are typically assumed which does not correspond to a bulk of existing knowledge (Lepparanta et al., 2015). In a recent study, (Tan et al., 2018) one possible direction to improve the simulation of ice-cover is shown. They demonstrated that including the conversion of snow to white or slush ice when the weight of ice and snow exceeds the buoyancy of the ice cover, can significantly improve the ice simulation results. In special case of saline lakes, effects of salts trapping in ice cover become important and should be adequately parameterized as well (Stepanenko et al., 2019).

7. (1) It would also be of interest to have a comparison of surface temperatures observed at 10cm to the model simulations to be sure that the daily cycle of temperature is well reproduced. This is a key feature for any further coupling to an atmospheric model. Could you add such a graph in the revised manuscript?

(2) The diurnal cycle of the surface temperature has a limited relevance to the problem of the vertical diffusion of gases originated from bottom. Below there is a graph showing the diurnal course of temperature for 1 month, July, averaged over all 5 years. There is a systematic difference between the results of models and observation data (e.g. up to 4 $^o$C for ALBM model), however, in this study we do not discuss the reason of these errors given the main focus of the study. (see attached figure: Fig.1)

8. (1) A comparison of methane profiles for ALBM and LAKE would also be interesting (climate change context, ...) even if no observation is available.

(2) The difference between these two models in methane simulation is significant: the maximum bottom concentration is 2.8 ppm and 1.1 ppm for ALBM and LAKE, respectively. The mean bottom methane concentration in the LAKE model (0.03 ppm) is an order of magnitude smaller than in the ALBM model (0.24 ppm). In general, the modeled concentration of methane in the lake is small, because of a high oxygen concentration. However, there are no observations available and we cannot consider one or another model as more realistic in this respect. Hence, we do not include this result into the main manuscript. (see attached figures: Fig.2, Fig.3)

**Technical comments**

1. (1) Page 4 line 12: change – by of

(2) We coud not find these corrections.

2. (1) Page 6 line 5: summarized (2)-(3) Page 6 line 5: changed to "summarized"

References:

1. Fer, I., Lemmin, U., Thorpe, S. A.: Cascading of water down the sloping sides of a deep lake in winter, Geophys. Res. Lett., 28(10), 2093-2096, https://doi.org/10.1029/2000GL012599, 2001.

2. Fer, I., Lemmin, U., Thorpe, S. A.: Winter cascading of cold water in Lake Geneva, J. Geophys. Res.-Oceans, 107(C6), 13-1, https://doi.org/10.1029/2001JC000828, 2002. Forbes G. S. and Meritt J. H.: Mesoscale vortices over

3. Kaden, H., Peeters, F., Lorke, A., Kipfer, R., Tomonaga, Y., Karabiyikoglu, M.: Impact of lake level change on deep-water renewal and oxic conditions in deep saline Lake Van, Turkey, Water Resour. Res., 46(11), https://doi.org/10.1029/2009WR008555, 2010.

4. Kirillin, G. B., Forrest, A. L., Graves, K. E., Fischer, A., Engelhardt, C., Laval, B. E.: Axisymmetric circulation driven by marginal heating in ice-covered lakes, Hydrol. Geophys. Res. Lett., 42(8), 2893-2900, https://doi.org/10.1002/2014GL062180, 2015.

5. Kiuru, P., Ojala, A., Mammarella, I., Heiskanen, J., Kämäräinen, M., Vesala, T., Huttula, T.: Effects of climate change on $CO_2$ concentration and efflux in a humic boreal lake: A modeling study, J. Geophys. Res.-Biogeo., 123, 2212– 2233, https://doi.org/10.1029/2018JG004585, 2018.

6. Lepparanta M. Freezing of lakes and the evolution of their ice cover: Springer Heidelberg New York Dordrecht London, 301 p.,doi:10.1007/978-3-642-29081-7, 2015.

7. Samolyubov, B. I.: Bottom stratified currents, Nauchny Mir, 464, 1999 (in Russian).

8. Stepanenko, V., Mammarella, I., Ojala, A., Miettinen, H., Lykosov, V., and Vesala, T.: LAKE 2.0: a model for temperature, methane, carbon dioxide and oxygen dynamics in lakes, Geosci. Model Dev., 9(5), 1977-2006, https://doi.org/10.5194/gmd-9-1977-2016, 2016.

9. V. M. Stepanenko, I. A. Repina, G. Ganbat, and G. Davaa. Numerical simulation of ice cover of saline lakes. Izv. Atmos. Ocean Phy.+,55(1):129–138, http://dx.doi.org/10.1134/S0001433819010092, 2019.

10. Tan, Z., Zhuang, Q., and Walter Anthony, K.: Modeling methane emissions from arctic lakes: Model development and site level study, J. Adv. Model Earth Sy., 7(2), 459-483, https://doi.org/10.1002/2014MS000344, 2015.

11. Tan, Z., Zhuang, Q., Shurpali, N. J., Marushchak, M. E., Biasi, C., Eugster, W., and Walter Anthony, K.: Modeling CO2 emissions from Arctic lakes: Model development and site-level study, J. Adv. Model Earth Sy., 9(5), 2190-2213, https://doi.org/10.1002/2017MS001028, 2017.

12. Tan, Z., Yao, H., Zhuang, Q.: A small temperate lake in the 21st century: Dynamics of water temperature, ice phenology, dissolved oxygen and chlorophyll-a. Water Resour. Res., 54, https://doi.org/10.1029/2017WR022334, 2018.

Additional comment: Attached file contains all changes in the manuscript and SI mentioned above.

Please also note the supplement to this comment:
https://www.hydrol-earth-syst-sci-discuss.net/hess-2019-146/hess-2019-146-AC2-supplement.zip

[Figure]

[Figure]

Surface temperature

**Fig. 1.** The diurnal cycle of the surface water temperature (in July), averaged over 5 years.

ALBM

LAKE

Methane concentration, *ppm*

**Fig. 2.** Vertical distribution of methane concentration in two models: ALBM, LAKE

[Figure]

**Fig. 3.** Vertical distribution of methane concentration (logarithmic scale) in two models: ALBM, LAKE

---

## Author Response (AR1)

Dear Referees and Editor,

we highly appreciate your time and work in revising our manuscript. We thank the Referees for the very important and helpful comments which helped us to find the mistakes and improve the modeling results. We went carefully through all the suggestions/comments/corrections and respond to them at our best.

Kind regards, Authors

———————————————

1. (1) Concerns arise however about the way one of the models - FLake - was treated in the study. The authors correctly state in the description of the model experiments that FLake "stands aside from the other 1-D models due to the . . . bulk-structure which employs the concept of <temperature profile> self-similarity. . . ". This high level of parameterization ensures computational efficiency of the model, which was primarily designed for prediction of surface temperatures in global/regional climate models and numerical weather prediction (NWP). On the other hand, the model parameterizations put some constraints on the model application to real lakes. One crucially important feature of FLake is that the model equations are derived in the assumption of preserving the heat capacity or volume of the lake. In this regard, the "baseline" configuration

applied in this study with the maximum lake depth as model input is inappropriate for FLake and would produce a priori incorrect results. Another issue of the parameterized model refers to correct choice of the few "shape factor" parameters determining the stratification pattern. The latter is not resolved in FLake numerically, but parameterized via few "shape factor"-constants related to the spatial integrals over the stratified layer. Several recent publications, including co-authorship of one of the authors of the present study, (Shatwell et al., 2016; Kirillin et al., 2017; Shatwell et al., 2019; Su et al., 2019) discussed the appropriate choice of the shape factor constants in FLake and have demonstrated that the set of constants used in the NWP-version of the model should be amended if the vertical thermal structure is in question apart from the lake surface temperatures. In particular, the unrealistically weak deep stratification and the corresponding high depth of the surface mixed layer, as reported in the present study, are the results of applying the NWP-constants together with the maximum lake depth as the model lake depth. Hence, the FLake-outcomes discussed here are useless or even misleading for the potential FLake-users. I see two ways of possible modification of the study: (i) excluding FLake from the study completely, confining the set of tested models to four models; (ii) re-designing the FLake-experiments in the corrected way with corresponding changes in the results/discussion.

(2) We generally agree with these suggestions of the referee and have re-designed experiments with FLake model testing the sensitivity of the model to one of the constants determining "shape factor" – dimensionless relaxation constant $C_{rc}$. In the current study, we have got the best agreement with the measured temperature profile using the value $C_{rc} = 0.3$ (in comparison with the standard value of 0.003) and chose this setting for the baseline experiment RefSim. The model now reproduces the temperature profile much better and we conducted other experiments including the one with mean lake depth and varying light attenuation coefficient with $C_{rc} = 0.3$.

(3) Page 7 line 15: The concept of the self-similarity in the FLake model (Sect. 2.2) includes the "shape factor" coefficient $C_\theta$, which determines the vertical temperature

profile below the mixed layer. Evolution of this parameter is controlled by a relaxation time scale $t_{rc}$ (Mironov, 2008). This time scale includes the dimensionless relaxation constant $C_{rc}$ having a default value of 0.003 in the model. This constant is a calibration parameter which is generally individual for each lake (Shatwell et al., 2016, Kirillin et al., 2017). We tested the sensitivity of the model to the variation of $C_{rc}$ using values 0.03, 0.3, 0.5, 1, 2, 3, 30 and achieved the best agreement with the measured temperature profile using $C_{rc} = 0.3$ (for details, see Sect. 3.1). In the following sections, this $C_{rc}$ setting is used in RefSim and other experiments with FLake model.

Page 10 line 11: In RefSim we test the model with different values of $C_{rc}$ (see Sect. 2.3 above). The best agreement with observed temperature profile we get using $C_{rc} = 0.3$. In particular, the error (RMSE$_c$) for surface temperature reduces from 3.4 ° C (using standard value of $c_{relax} = 0.003$) to 2.5 ° and becomes close to other models. At the depth of $\sim$ 6-10 m RMSE$_c$ reduces from 5.7 ° C (using standard value of $c_{relax} = 0.003$) to 5.2 °. In addition, RMSE$_c$ reduces from 3.3-0.7 ° C to 1.8-0.2 ° C at the depths 20-27 m.

We added new figures with new results for FLake model to the main manuscript: Fig. 2, Fig. 3, Fig. 4 ; add new figures to SI: Fig. S2; changed metrics for FLake model performance in Tables: S2, S3, S4.

2. (1) Regarding the biogeochemical part, the study would benefit from an extension of the discussion on possible ways of improving the representation of biogeochemistry, in particular, the deep oxygen (chlorophyll) maximum in oligotrophic lakes, and the vertical distribution of carbon dioxide across the water column.

(2)-(3) Page 19 line 34: Successful representation of maximal $O_2$ content below mixed layer during summer may be an important modelling skill for simulating correctly $CH_4$ in a lake because the former acts as a sink region for the latter. Furthermore, realistic oxygen concentration reduction during periods of stable stratification means that the respective $CO_2$ production from aerobic organic matter decomposition is reproduced

reasonably as well.

Page 21 line 21: As stated in the previous section, a realistic decay of oxygen content during stratified periods in ALBM and LAKE models suggests that $CO_2$ amount produced by aerobic decomposition of organic matter both in water column and in the top part of sediments is reasonably simulated as well. Satisfactory agreement of computed oxygen in the mixed layer and below with observations implies that photosynthesis minus respiration rate is fairly captured as well, and so do the models for $CO_2$ gain or loss due to these processes. Hence, the primary drawback of the models used in respect to DIC simulation is likely to be not explicitly simulating transport of carbon species from catchment to a water body. Thus, modelling approaches coupling the catchment and a lake presented recently (Futter et al., 2008; Duffy et al., 2018; McCullough et al., 2018) should be elaborated and wider used.

3. (1) Note also that the temperature profile within the ice cover in FLake is not assumed to be linear, but parameterized via a time-varying shape-function with a linear asymptotic.

(2) Yes, that is correct.

(3) Page 5 line 20: In FLake model, the temperature profile within the ice cover is parameterized via a time-varying shape-function having a linear asymptotic.

References: 1. Duffy C.J., Dugan H.A., Hanson P.C.: The age of water and carbon in lake-catchments: A simple dynamical model, Limnol. Oceanogr. Lett.,Vol. 3, No. 3, pp. 236–245, https://doi.org/10.1002/lol2.10070, 2018.

2. Futter M.N., Starr M., Forsius M., Holmberg M.: Modelling the effects of climate on long-term patterns of dissolved organic carbon concentrations in the surface waters of a boreal catchment, Hydrol. Earth Syst. Sci., Vol. 12, No. 2, pp. 437–447, https://doi.org/10.5194/hess-12-437-2008, 2008.

3. Kirillin, G., Wen, L., Shatwell, T.: Seasonal thermal regime and climatic trends

in lakes of the Tibetan highlands, Hydrol. Earth Syst. Sci.,21(4), 1895-1909, https://doi.org/10.5194/hess-21-1895-2017, 2017.

4. McCullough I.M., Dugan H.A., Farrell K.J., Morales-Williams A.M., Ouyang Z., Roberts D., Scordo F., Bartlett S.L., Burke S.M., DoubekJ.P., Krivak-Tetley F.E., Skaff N.K., Summers J.C., Weathers K.C., Hanson P.C.: Dynamic modeling of organiccarbon fates in lake ecosystems, Ecol. Modell. Vol. 386. pp. 71–82, https://doi.org/10.1016/j.ecolmodel.2018.08.009, 2018.

5. Mironov, D. V.: Parameterisation of lakes in numerical weather prediction. Description of a lake model, COSMO Technical Report 11,Deutscher Wetterdienst, Offenbach am Main, Germany, 2008. 6. Shatwell, T., Adrian, R., and Kirillin, G.: Planktonic events may cause polymictic-dimictic regime shifts in temperate lakes., Sci. Rep.-UK,256, 24361, https://doi.org/10.1038/srep24361, 2016.

Additional comment: Attached file contains all changes in the manuscript and SI mentioned above.

Please also note the supplement to this comment: https://www.hydrol-earth-syst-sci-discuss.net/hess-2019-146/hess-2019-146-AC1-supplement.zip

[Figure]

Hydrol. Earth Syst. Sci. Discuss.,
https://doi.org/10.5194/hess-2019-146-AC2, 2019

1.  (1) It is stated that the study is a continuity of LakeMIP exercises accounting for biogeochemical processes comparisons. Although the first part of the paper is dedicated to thermal stratification and ice cover study, and involves the five models, the fact that only two of them have the possibility to simulate O2 and CO2 dynamics, indicate that this is not a real LakeMIP type exercise, and therefore constitutes a limitation in my view to be considered as a true intercomparison model experiment for vertical gaz transfer.

(2) Dissolved gas transport throughout the water column is mainly carried out by turbu-

lence. Despite the fact that only two of the models have an ability directly to reproduce the concentration the gases (oxygen and carbon dioxide) our numerical experiments with a passive tracer (which can be seen as a "prototype" of gas or other constituent) involving four models out of five showed the potential simulation of gases governed by seasonal stratification and ice cover. We can consider this experiment as the inter-comparison model experiment in respect to primary physical controls of gas dynamics. We consider the intercomparison experiment involving full biogeochemical models as a next step of this study.

(3) No comments/corrections are added to the manuscript.

2. (1) In the first LakeMIP exercise (Stepanenko et al., 2010) the sensitivity of lake depth has been studied and the experiment setup accounted for simulations with maximum depth, local depth and average depth. And this is crucial especially (maybe only) for FLake model as it was demonstrated that an average lake depth was necessary for FLake simulation in order to be conservative in terms of energy. In the current setup, the maximum lake depth is used for all models and this is contradictory with a correct use of FLake which should be run using the LDSim configuration. It could be interesting to compare in the same graph RefSim and LDSim at least for FLake.

(2) The main focus of our study is the vertical diffusion which is the main driver for the gas distribution. Of major interest are such gases as carbon dioxide and methane which are primarily produced in lake sediments. Setting the maximum lake depth becomes crucial in terms of the total vertical distance required for transport of bottom-originated gases to the atmosphere. For this reason, we conduct the baseline experiment with the maximum depth. We added new figures to the manuscript with results from the LDSim experiment – FLake showed the largest sensitivity to the variation of the lake depth among the other models: e.g. $RMSE_c$ for surface temperature as well as for the whole temperature profile reduced almost twofold (see Table S4, Fig. S3, S4). This suggests potential limitation of FLake for correctly simulating gas dynamics in deep lakes (given the biogeochemical block is added to the model).

[Figure]

Interactive
comment

(3) Page 10 line 24: Remarkably, FLake demonstrates the largest sensitivity among all models to the lake depth – $RMSE_c$ is reduced approximately twofold (from 2.5 ° C to 1.28 ° C) (see SI, Table S4, Fig. S3, S4) when using a mean depth of 13.32 m instead of the maximal depth 37.5 m. The thickness of mixed layer depth reduces up to 4.5 m and have a better agreement with the observations in comparison with RefSim experiment. We added the mean depth of the mixed layer for all models in the LDSim experiment to the Table S2 in SI. We also put two additional figures for LDSim simulation to SI: Fig. S3, S4.

3. (1) As a consequence, the sensitivity test on light extinction for FLake is not relevant since the thermal profile cannot be well simulated, and therefore should be conducted with a depth of 13.32 m.

(2) We redid all FLake simulations using calibrated shape factor, and temperature profile is now much better reproduced compared to results presented in the first version of the manuscript. Correspondingly, the sensitivity test on light extinction coefficient with FLake demonstrates now result similar to other models (see new Fig. 4).

(3) Page 11 line 32: This leads to temperature change at respective depths up to 4 $°C$ in FLake model and 8 $°C$ in ALBM and LAKEoneD. Fig. 4 contains only 3 models with different types of turbulent closure because the effect of varying the light attenuation coefficient is very similar in models on the same type (e.g. k-$\epsilon$ models LAKEoneD and LAKE).

We also added the result of ExtMaxSim simulation for FLake in Fig. 4 in the manuscript.

**Specific comments**

4. (1) Looking at the bathymetry indicates strong gradients from the shoreline to the point of maximum depth. There are probably 3d-circulations that take place when dense waters flow along the bottom slopes. And these circulations are not accounted for in the 1d simulations. What is in your view the potential impact on these circulations

on the thermal stratification and ice cover? And on modeled vertical transport? It would be interesting to add a discussion on that particular point.

(2) 1D models have certain limitations when it comes to such a 3-dimensional system as lake. They do not capture all the lake mixing mechanisms such as density driven currents, which can be important under certain conditions. In general, these currents are very slow (only a few cm s$^{-1}$) (Bengtsson, 2012) and are not important in terms of the turbulence production. The strong contribution of these flows in the vertical exchange of deep and near-surface water was found for deep lakes (> 100 m) with an extended literal zone, e.g. Lake Geneva and Lake Van (Fer et al., 2001, Fer et al. 2002, Kaden et al., 2010). For Harp Lake, in particular, this process may not be important during summer due to a smaller depth and a limited area of shallow part. In winter, a large-scale convective circulation (up to 3-5 cm s$^{-1}$) due to heat exchange at the sediment-water interface may develop under ice (Kirillin et al., 2015). However, the importance of this circulation in terms of gas transfer have not been studied yet. At the moment, this type of the lake circulation is a topic of the ongoing research and it is not included into 1D models. Modeling such 3-d phenomena requires an application of 2D or 3D models.

(3) Added to Page 24 line 22 : - 1D models have certain limitations when applying to such a 3-dimensional system as lake. They do not capture all the lake mixing mechanisms such as density driven currents, which can be important under certain conditions (Samolyubov,1999). It was found that for deep lakes with extended literal zone, such as Lake Geneva (Fer et al., 2001, 2002) and Lake Van (Kaden et al., 2010), these flows can be significant in terms of the vertical heat and gas transfer. In particular, for Harp Lake it may not be important due to a smaller depth and a not large shallow area. In winter, a large-scale convective circulation (up to 3-5 cm s$^{-1}$) due to heat exchange at the sediment-water interface may develop under ice (Kirillin et al., 2015). However, the importance of this circulation in terms of gas transfer have not been studied yet. So far, to the best of our knowledge, these currents are not parameterized in 1D models.

[Figure]

5. (1) The abstract mentions the need to improve biogeochemical processes in lake models to enhance weather prediction and climate projection capabilities. I'm not convinced improving biogeochemical processes will improve weather prediction. What is crucial in weather prediction or climate modeling is to simulate a correct surface temperature and fluxes because these are the variables that will be used in the coupling to the atmosphere. That's true that in climate simulations the knowledge of carbon dioxide or methane emissions are of high interest, however to my knowledge only the LAKE model offers the capability to simulate CO2, O2 and CH4 dynamics and be coupled to a climate model. Please add a discussion on that point to also highlight the difficulty to increase the complexity of lake models and ensure a correct coupling to an atmospheric climate model.

(2)-(3) As in this study we do not focus on methane, due to absence of respective measurement data, and discuss models performance for $CO_2$ content, we added a comment on issues which will arise when implementing lacustrine $CO_2$ emissions in climate models (Page 21 line 29): The important role of catchment processes in building up the DIC levels in lakes introduces an extra difficulty for implementing lacustrine $CO_2$ emissions in the Earth system models. This is caused by the necessity to provide regional or global data on lake's catchments geometric, physical and biogeochemical properties, which are not currently available. In relation to this, we can also note a faster progress on a roadmap to introduce lake $CH_4$ dynamics in climate models, where simulations passed from site-levels studies to regional estimates (Tan et al., 2015), whereas $CO_2$ modeling is currently confined to individual lakes (Stepanenko et al., 2016; Tan et al., 2017; Kiuru et al., 2018).

6. (1) Ice cover is a key variable for vertical transfer of gases. It has been shown that freezeup or brake-up presented delays of several weeks potentially. Don't you think more effort should be put on the representation of ice and snow over ice in lake models, especially when working in NWP and/or climate contexts?

(2)-(3) We agree and added the following comment at Page 23 line 18: This implies

more efforts of a community should be spent on elaborating ice-snow schemes in lake models, where currently vertical homogeneity and temporal invariance of optical and thermodynamic properties are typically assumed which does not correspond to a bulk of existing knowledge (Lepparanta et al., 2015). In a recent study, (Tan et al., 2018) one possible direction to improve the simulation of ice-cover is shown. They demonstrated that including the conversion of snow to white or slush ice when the weight of ice and snow exceeds the buoyancy of the ice cover, can significantly improve the ice simulation results. In special case of saline lakes, effects of salts trapping in ice cover become important and should be adequately parameterized as well (Stepanenko et al., 2019).

7. (1) It would also be of interest to have a comparison of surface temperatures observed at 10cm to the model simulations to be sure that the daily cycle of temperature is well reproduced. This is a key feature for any further coupling to an atmospheric model. Could you add such a graph in the revised manuscript?

(2) The diurnal cycle of the surface temperature has a limited relevance to the problem of the vertical diffusion of gases originated from bottom. Below there is a graph showing the diurnal course of temperature for 1 month, July, averaged over all 5 years. There is a systematic difference between the results of models and observation data (e.g. up to 4 $^o$C for ALBM model), however, in this study we do not discuss the reason of these errors given the main focus of the study. (see attached figure: Fig.1)

8. (1) A comparison of methane profiles for ALBM and LAKE would also be interesting (climate change context, ...) even if no observation is available.

(2) The difference between these two models in methane simulation is significant: the maximum bottom concentration is 2.8 ppm and 1.1 ppm for ALBM and LAKE, respectively. The mean bottom methane concentration in the LAKE model (0.03 ppm) is an order of magnitude smaller than in the ALBM model (0.24 ppm). In general, the modeled concentration of methane in the lake is small, because of a high oxygen con-

centration. However, there are no observations available and we cannot consider one or another model as more realistic in this respect. Hence, we do not include this result into the main manuscript. (see attached figures: Fig.2, Fig.3)

**Technical comments**

1. (1) Page 4 line 12: change – by of

(2) We coud not find these corrections.

2. (1) Page 6 line 5: summarized (2)-(3) Page 6 line 5: changed to "summarized"

[Figure]

[Figure]

**Surface temperature**

T° C

hour

ALBM — LAKE — LAKEoneD — MTCR-1 — Observed
FLake

**Fig. 1.** The diurnal cycle of the surface water temperature (in July), averaged over 5 years.

[Figure]

ALBM

LAKE

Methane concentration, $ppm$

**Fig. 2.** Vertical distribution of methane concentration in two models: ALBM, LAKE

[Figure]

[Figure]

**Fig. 3.** Vertical distribution of methane concentration (logarithmic scale) in two models: ALBM, LAKE

[Figure]

**List of additional changes**

1. SI Page 5 Table S4: correction of two numbers for model ALBM for LDSim (typo in the first version)

2. Page 10 line 16: text for FLake model analysis was rearranged .

[revised manuscript text omitted]

**Supporting information to the paper by**
**Guseva, S., Bleninger, T.,Jöhnk, K., Polli, B.A., Tan, Z., Thiery, W., Zhuang, Q., Rusak, J., Yao, H., Lorke, A. and Stepanenko, V. "Multimodel simulation of vertical gas transfer in a temperate lake"**

5 **1 Figures**

[Figure]

**Figure S1.** Time-series of meteorological variables: air temperature, wind speed and precipitation. Black line indicates the moving average with a window of half a month.

[Figure]

**Figure S2.** Taylor diagram for the water temperature at (a) surface (b) all depths in RefSim. Grey lines indicate $RMSE_c$ ($^{\circ}C$), the radial distance from the origin is the standard deviation ($^{\circ}C$), an azimutal position denote the correlation coefficient (Taylor et al., 2001).

[Figure]

**Figure S3.** Time-depth pattern of temperature in Harp Lake (14.07.2010-19.10.2015), LDSim and observed data.

[Figure]

**Figure S4.** Taylor diagram for the water temperature at (a) surface (b) all depths in LDSim. Grey lines indicate $RMSE_c$ ($^\circ C$), the radial distance from the origin is the standard deviation ($^\circ C$), an azimutal position denote the correlation coefficient.

[Figure]

**Figure S5.** Time-series of the surface flux, normalized by bottom flux, and the total tracer amount in the water column: (a, b) RefSim, 37.5 m, (c, d) LDSim, 13.32 m.

[Figure]

**Figure S6.** Taylor diagram for the modeled $O_2$ concentration at depths of (a) 1 m (b) 18 m in GasSim.

**2 Tables**

**Table S1.** Biogeochemical processes represented in ALBM, LAKE models.

|  | ALBM | LAKE |
|---|---|---|
| Gas exchange at the air-water interface | Surface renewal model (Heiskanen et al., 2014) | Surface renewal model (MacIntyre et al., 2010; Heiskanen et al., 2014) |
| Sedimentary oxygen demand | (Hanson et al., 2004; Stefan and Fang, 1994) | (Walker and Snodgrass, 1986) |
| Photosynthesis | (Tian, 2006; Hipsey et al., 2008; Li et al., 2010) | (Megard et al., 1984; Stefan and Fang, 1994) |
| Respiration | (Hanson et al., 2004) | (Stefan and Fang, 1994) |
| Biochemical oxygen demand | (Hipsey et al., 2008; Hanson et al., 2011) | (Stefan and Fang, 1994) |
| DIC from inflow | (Hanson et al., 2004) | - |
| DOC and POC | (Hanson et al., 2004) | - |

**Table S2.** Mean depth of mixed layer in models in RefSim and LDSim.

| Mean depth of mixed layer, m [1] | ALBM | FLake | LAKE | LAKEoneD | MTCR-1 | Observed |
|---|---|---|---|---|---|---|
| RefSim | 2.4 | 5.6 | 4.8 | 3.9 | 5.9 | 3.6 |
| LDSim | 2.1 | 4.5 | 4.9 | 3.7 | 5.8 | 3.6 |

a. [1] is calculated as a mean depth of a maximum for the buoyancy frequency: $N^2 = \dfrac{g}{\rho_0}\dfrac{\partial \rho(T)}{\partial z}$.

b. The period of averaging is starting from spring ice-off and ending at August, 31

**Table S3.** Dates of ice formation and ice melt in RefSim.

|  | Ice- | ALBM | LAKE | LAKEoneD | MTCR-1 | FLake | Observations |
|---|---|---|---|---|---|---|---|
| **2010-2011** | cover | 24.12.2010** | 15.12.2010 | 06.12.2010 | 05.01.2011** | 22.01.2011** | 10.12.2010 |
|  | off | 01.05.2011 | 03.04.2011 | 02.05.2011 | 04.05.2011* | 09.04.2011 ** | 27.04.2011 |
| **2011-2012** | cover | 24.12.2011 | **28.12.2011** | **10.12.2011*** | 29.12.2011 | 15.01.2012 ** | 28.12.2011 |
|  | off | 02.04.2012 | 12.03.2012** | 23.03.2012 | 25.03.2012 | 21.03.2012 * | 30.03.2012 |
| **2012-2013** | cover | 05.01.2013** | 27.12.2012** | **28.11.2012*** | 24.12.2012* | 23.01.2013** | 12.12.2012 |
|  | off | 22.04.2013 | 29.03.2013** | 01.05.2013 | 04.05.2013* | 26.03.2013** | 25.04.2013 |
| **2013-2014** | cover | 13.12.2013* | 12.12.2013* | 10.12.2013* | 13.12.2013* | 01.01.2014** | 01.12.2013 |
|  | off | 29.04.2014 | 14.04.2014** | 11.05.2014* | 13.05.2014* | 21.04.2014* | 30.04.2014 |
| **2014-2015** | cover | 29.12.2014** | 22.12.2014** | 17.11.2014** | 12.12.2014* | 05.01.2015** | 02.12.2014 |
|  | off | 17.04.2015 | 02.04.2015** | 04.05.2015** | 07.05.2015** | 15.04.2015 | 21.04.2015 |

a. Difference between the model and observations:* > 1 week ; ** > 2 weeks

b. **10.12.2011** dates denoted by bold indicate multiple freezing-melting events after the date

**Table S4.** Statistics of the surface temperature of the lake in RefSim, LDSim, ExtMinSim, ExtMaxSim.

| Experiment | | ALBM | FLake | LAKE | LAKEoneD | MTCR-1 |
|---|---|---|---|---|---|---|
| RefSim | $h_{max} = 37.5$ m | 2.13 ; | 0.36 ; | 1.11 ; | 0.15 ; | 0.68 ; |
| | DM (°C) ; RMSE$_c$ (°C) | 2.17 | 2.5 | 1.52 | 1.32 | 2.62 |
| LDSim | $h_{ave} = 13.32$ m | 1.1 ; | 0.28 ; | 1.03 ; | 0.21 ; | 0.16 ; |
| | DM (°C) ; RMSE$_c$ (°C) | 3.5 | 1.28 | 1.51 | 1.25 | 2.15 |
| ExtMinSim | µ= 0.28 m$^{-1}$ | 2.08 ; | 0.37 ; | 1.15 ; | 0.08 ; | 0.7 ; |
| | DM (°C) ; RMSE$_c$ (°C) | 2.12 | 2.7 | 1.68 | 1.74 | 3.07 |
| ExtMaxSim | µ= 0.68 m$^{-1}$ | 2.1 ; | 0.38 ; | 1.13 ; | 0.2 ; | 0.7 ; |
| | DM (°C) ; RMSE$_c$ (°C) | 2.25 | 2.48 | 1.51 | 1.02 | 2.03 |

**3 Appendix**

**Brief description of field data at Harp Lake, Ontario, Canada**

Oct. 8, 2015 (by Huaxia Yao)

**1. Site and lake facts**

**Table S5.** Chemical status of lake water.

|  | mg L$^{-1}$ |  | mg L$^{-1}$ |  | mg L$^{-1}$ |
|---|---|---|---|---|---|
| Salinity | 15 | DOC | 1.8 | DIC | 1.5 |
| Cl | 2.8 | TP | 4.4 | NO$_3$ | 0.1 |
| SO$_4$ | 6.6 | Ca | 2.6 |  |  |
| silica | 1.8 | Na | 2.2 |  |  |

5    **2. High frequency data from a raft buoy** (10 minute intervals)

**Table S6.** Meteorological measurements.

|  | Height, m | Period of measurements | Instrument |
|---|---|---|---|
| Air temperature | 1.75 | July 14, 2010 – present | Vaisala HMP45C air temperature and relative humidity sensor |
| Relative humidity | 1.75 | July 14, 2010 – present | Vaisala HMP45C air temperature and relative humidity sensor |
| Wind speed | 2 | July 14, 2010 – present | RM Young wind monitor model 05103 |
| Wind direction | 2 | July 14, 2010 – present | RM Young wind monitor model 05103 |
| Atmospheric longwave radiation (earlier data can be got from another land-based station) |  | Nov. 5, 2012 – present | Kipp & Zonen CGR3 Pyrgeometer |
| Short-wave radiation (earlier data can be got from another land-based station) |  | Nov. 5, 2012 – present | Kipp & Zonen CGR3 Pyrgeometer |
| Precipitation |  |  | not measured on the raft, but available from neighboring station |
| Atmospheric pressure |  |  | not measured on the raft, but available from neighboring station |

**Table S7.** Measurements in lake water.

|  | Depth, m | Period of measurements | Instrument |
|---|---|---|---|
| Chlorophyll-a | 1 | Sept. 7, 2011 –present | YSI 6025 Chlorophyll sensor |
| $CO_2$ concentration | 0.39 | Mar. 12, 2012 –present | Vaisala GMP343 $CO_2$ Probe with sintered PTFE filter |
| Electrical conductivity | 1, 18 | Sept. 11, 2011 –present | YSI 6560 temperature and conductivity sensor |
| DO concentration | 1, 18 | June 30, 2011 – present | YSI 6150 optical oxygen sensor |
| Water temperature | 0.1 to 10.1 (every 0.25 m intervals between 0.1 to 10.1 m) | July 14, 2010 –present | PME thermistor T-chain with pressure transducer |
| Water temperature | 0.1 to 27.1 (1m interval) | Mar. 12, 2012 - present | Campbell Scientific custom thermistor string of model 109 temperature probe |

**3. Low frequency lake data (regular monitoring)** (weekly or bi-weekly for ice-free seasons)

Lake profile data: temperature and DO, at 1m interval from 0.1 m to 35m. Started late 1970s. For our selected years 2010-2015, these weekly profile measurements do not cover the winter or ice seasons. Secchi disk depth (bi-weekly or monthly): since 1978

Ice date: 1978 to 2015 Ice thickness (monthly): 1978-1993; and 2013-2015

5 Regular meteorology data (daily or hourly): land-based stations, 1978-2015.

**4. Hydrology data**

Stream flow discharge (daily): at 5 inlet streams (HP3, HP3a, HP4, HP5, HP6a), and the outlet (HP0), long-term data Stream water chemistry (weekly or bi-weekly): at all streams

**5. Gap filling, QA/QC**

10 The data are basically from raw data of the raft/buoy sensors, with a simple quality check and gap filling.

For our LMIP project, the data have been edited and gap-filled. Missing data (gaps) in atmospheric forcing are infilled mainly by using another dataset of a neighbouring climate station (backyard of DESC office site). Except for air temperature, any missed hourly data at the raft is provided by the values at the neighbouring station. The missings in air temperature are filled by regressions between the two stations which are established for each month (using available hourly data). Any doubt and concern with data are kindly reported to participants and

15 Huaxia Yao for a solution.